# The rhythmic coupling of Egr-1 and Cidea regulates age-related metabolic dysfunction in the liver of male mice

Jing Wu [1,2,3,5], Dandan Bu[1,5], Haiquan Wang[1], Di Shen [1], Danyang Chong[1], Tongyu Zhang[1], Weiwei Tao[2], Mengfei Zhao[1], Yue Zhao [1], Lei Fang [1], Peng Li [4] ✉, Bin Xue [3] ✉ & Chao-Jun Li [2] ✉

The liver lipid metabolism of older individuals canbecome impaired and the circadian rhythm of genes involved in lipid metabolism is also disturbed. Although the link between metabolism and circadian rhythms is already recognized, how these processes are decoupled in liver during aging is still largely unknown. Here, we show that the circadian rhythm for the transcription factor Egr-1 expression is shifted forward with age in male mice. *Egr-1* deletion accelerates liver age-related metabolic dysfunction, which associates with increased triglyceride accumulation, disruption of the opposite rhythmic coupling of Egr-1 and Cidea (Cell Death Inducing DFFA Like Effector A) at the transcriptional level and large lipid droplet formation. Importantly, adjustment of the central clock with light via a 4-hour forward shift in 6-month-old mice, leads to recovery the rhythm shift of Egr-1 during aging and largely ameliorated liver metabolic dysfunction. All our collected data suggest that liver Egr-1 might integrate the central and peripheral rhythms and regulate metabolic homeostasis in the liver.

Aging is associated with metabolic disorders such as obesity, hyperglycemia, and hyperlipidemia[1] resulting from age-related nutrient sensing deregulation, mitochondrial dysfunction, cellular signaling pathway reprogramming, and insulin resistance[2]. As the central metabolic organ, the liver exhibits signs of progressive metabolic disorder during aging, such as triglyceride (TG) accumulation enhancement, fatty acid oxidation inhibition, and lipolysis impairment[3–5]. On the other hand, the circadian rhythms in the superchiasmatic nucleus (SCN) and peripheral tissues also undergo significant changes during aging[6]. The rhythm of Per1 expression seen in the lungs of young rats (2 months old) is absent in old animals (24–26 months old)[7]. The expression amplitude of clock genes such as *Clk* and *Pdp1cε* is also decreased in the head and body tissues of aged

flies (58 days old)[8]. These declines in peripheral clocks may affect rhythmic changes in hormone release, temperature regulation, and metabolism[6].

Many metabolic processes in the liver, such as gluconeogenesis, lipogenesis, and bile acid synthesis, also show rhythmic changes regulated by master circadian rhythms, and these metabolic processes exhibit progressive alterations with age[6,9,10]. It has been reported that 44.8% of genes that are rhythmically expressed in young mice exhibit rhythm disorders in elderly mice; these genes are mainly involved in glycerol metabolism and sterol metabolism[2]. Deficiencies in circadian genes, such as *Bmal1* and *Per1/2*, can accelerate the aging process[11]. Thus, impairment of metabolic and circadian rhythmic synchronization might be particularly important with

[1]Ministry of Education Key Laboratory of Model Animal for Disease Study, Model Animal Research Center of the Medical School, Nanjing University, Nanjing 210093 Jiangsu Province, China. [2]State Key Laboratory of Reproductive Medicine and China International Joint Research Center on Environment and Human Health, Center for Global Health, School of Public Health, Nanjing Medical University, Nanjing 211166, China. [3]Core Laboratory, Sir Run Run Hospital, Nanjing Medical University, Nanjing 211166 Jiangsu, China. [4]Institute of Metabolism & Integrative Biology (IMIB), Fudan University, Shanghai 200438, China. [5]These authors contributed equally: Jing Wu, Dandan Bu. ✉e-mail: peng_li@fudan.edu.cn; xuebin@njmu.edu.cn; lichaojun@njmu.edu.cn

age increased in the liver, but how the processes are coordinated still needs to be explored.

Early growth response-1 (Egr-1) is a member of the immediate early gene family that recognizes a highly conserved GC-based promoter sequence and then regulates the expression of many target genes[12,13]. Egr-1 can be activated by a variety of stimuli, including cytokines, growth factors, and hormones, and regulates cell proliferation, cell metabolism, and the hepatic clock circuitry[14–16]. It has been reported that both the mRNA and protein levels of Egr-1 are significantly decreased in senescent cell lines[17,18]. RNA sequencing (RNA-seq) data have shown that Egr-1 expression in the livers of 21-month-old mice is significantly decreased[19]. Previous studies have also shown that Egr-1 can regulate glucose and lipid metabolism and act as a molecular brake that prevents excessive stimulation and regulates fluctuating blood glucose levels under physiologic conditions[20,21]. After a meal, Egr-1 can be induced by insulin in skeletal muscle cells and inhibit insulin receptor phosphorylation, thus reducing insulin sensitivity[20]. After long-term fasting, Egr-1 can be activated by glucagon and regulate the expression of gluconeogenesis genes in the liver[21]. Moreover, Egr-1 acts as an important regulator in lipid metabolism. Egr-1 enhanced by insulin directly inhibits patatin-like phospholipase domain containing 2 (ATGL) transcription and inhibits lipolysis of adipocytes[22]. Egr-1 also affects the transcription of the key cholesterol synthesis genes *Hmgcr*, *Cyp51*, *Me1*, and *Sqle* and promotes cholesterol anabolism[23]. In addition, our previous work has demonstrated that Egr-1 is rhythmically expressed in the livers of young mice and is required for the circadian expression patterns of several core clock genes in the liver; it especially regulates the transcriptional activity of the biorhythm gene *Per1*. On the other hand, the rhythm of Egr-1 is also regulated by BMAL1/CLOCK heterodimer expression[16]. Thus, we speculated that Egr-1 may act as a mediator to regulate the age-associated cooperation between circadian rhythms and metabolic patterns.

In this work, we demonstrate that the zeitgeber time (ZT) of Egr-1 peak expression is shifted forward with age. *Egr-1* deletion accelerates liver age-related TG accumulation by enhancing CD36 expression to facilitate fatty acid uptake and enhancing Cidea (cell death inducing DFFA like effector A) transcriptional expression to form large lipid droplets. The rhythmic coupling of Egr-1 and Cidea can regulate the formation of large lipid droplets in a BMAL1/CLOCK-dependent manner. Aging disrupts the coupling between Egr-1 and Cidea and facilitates large lipid droplet formation, resulting in age-related metabolic dysfunction of the liver. These results indicate that Egr-1 is a key mediator that regulates the age-associated cooperation between circadian rhythm and lipid metabolism in the liver.

## Results

### Liver rhythmic lipid metabolism is disrupted with age increased
To determine how lipid metabolism changes with age increased, we detected triglyceride (TG) accumulation and the results suggested that TG levels in the liver increased with age (Fig. 1A). Hematoxylin and eosin (H&E) staining and Oil Red O staining also showed that lipid droplet accumulation in the livers of mice increased at 12 months (Fig. 1B), when positive staining for the aging marker β-galactosidase (β-gal) began, and was significantly increased at 21 months (Fig. 1C). This observation indicated a phenomenon of metabolic dysfunction in the liver with age increase. Moreover, to determine how rhythmic metabolism and the hepatic clock change with age increased, mice of different ages (2, 6, and 12 months of age) were sacrificed every 4 h started at zeitgeber time (ZT) 1 over the circadian cycle (ZT1, 5, 9, 13, 17, 21) and hepatic transcriptomic analysis was performed. Heatmaps and Venn diagrams display circadian genes selected using the nonparametric algorithm JTK_cycle[24] and genes with p. adjust<0.05 were regarded as circadian genes. Exclusive circadian genes were found and displayed by heatmaps in the 2, 6, and 12-month groups (Fig. 1D–F).

Transcriptomics revealed 621 genes exclusively oscillatory in 2-month group, 579 genes exclusively oscillatory in the 6-month group, and 428 genes were only rhythmic in the 12-month group. Among the three groups, only 12 genes kept similar rhythms and fewer metabolism-related genes maintained consistent rhythms (Fig. 1G). GO enrichment analysis of oscillatory genes in a 2-month group and selecting the top 20 biological pathways indicated that 45.1 percent of these processes were related to lipid metabolism (Fig. 1H). However, with age increased, the proportion of lipid metabolism-related pathways gradually decreased to 28.3 percent in a 6-month group (Fig. 1I) and only 3 percent in 12-month group (Fig. 1J). Thus, these results revealed that rhythmic lipid metabolism was disordered with aging.

Detailed analysis of the circadian rhythm changes from hepatic transcriptomic and RT-PCR confirmed the rhythmicity change of most circadian genes in 2, 6, and 12 months of age livers. It is intrinsic that although the phase of most circadian genes in 6 months of age livers was moved forward or backward compared with 2 months of age livers, it was similar in 2 and 12 months of age livers, only the amplitude of *Bmal1* and *Clock* and *Rorα* was significantly changed (Supplementary Fig. 1A–I). However, the related gene number in enriched circadian rhythm-related biological pathways was remarkably decreased in 12-month-old mice except for the entrainment of the circadian clock and the entrainment of the circadian clock by photoperiod. And some pathways even disappeared like circadian sleep/wake cycle non-REM sleep, positive regulation of circadian sleep/wake cycle, and negative/positive regulation of circadian rhythm, which indicated that sleep/wake problems emerged in old age (Supplementary Fig. 1J). Meanwhile, the regulation of core circadian gene on TG accumulation is totally different between 2- and 12-month-old mice, in which knockdown of Bmal1 or Clock or Rorα could enhance TG accumulation in 2-month-old mice[25–27], while decreased TG accumulation in 12-month-old mice (Supplementary Fig. 1K–M), which suggested the "normal" 2-month-old relationship between core clock genes and lipid metabolism was disrupted in the liver of 12-month-old mice.

### The rhythmic phase of Egr-1 in the liver shifts forward with aging
In order to figure out the regulators to connect the circadian rhythms and metabolic patterns with age increased, we further analyzed the above transcriptomic data. The heatmap vividly displayed that core clock genes and clock-controlled genes (CCGs) in the circadian rhythm process had different rhythms in different age groups (Fig. 2A). We found that the rhythm of early-response transcription factor Egr-1 in multiple circadian rhythm processes was changed, which have been reported to participate in lipid metabolism[22]. We then detected the circadian expression of Egr-1 and found that the rhythmicity shifted with age. The mRNA expression of Egr-1 in the livers of 2-month-old mice peaked at ZT5 and declined thereafter, showing the lowest expression at ZT17. Interestingly, with increasing age, the peak of Egr-1 mRNA expression advanced, occurring at ZT1 in 6-month-old mice and at ZT21 in 12-month-old mice (Fig. 2B). The protein expression levels of Egr-1 in the livers of 2-month-old mice peaked at ZT13 and then advanced to ZT9 in 6-month-old mice and to ZT5 in 12-month-old mice (Fig. 2C-F). Moreover, by analyzing gene array data (GSE57809) of live young and old mice[19], we found that the mRNA levels of *Egr-1* were markedly decreased in old mice (Supplementary Fig. 2A). We also detected the protein levels of Egr-1 at ZT5 and further confirmed their slight decline in the livers of 21-month-old mice (Supplementary Fig. 2B, C). Moreover, we find an increasing tendency in Egr-1 expression in the livers of 6-month-old mice (Supplementary Fig. 2D, E). These results indicate that the circadian rhythm of liver Egr-1 is also altered with aging.

To examine whether the changed rhythm of liver Egr-1 was relevant to the metabolic dysfunction with age increase, we compared genes in the lipid-related biological process of Fig. 1H–J with the top 2000 genes from a published dataset Egr-1 ChIP-seq (GSM1037682)[28],

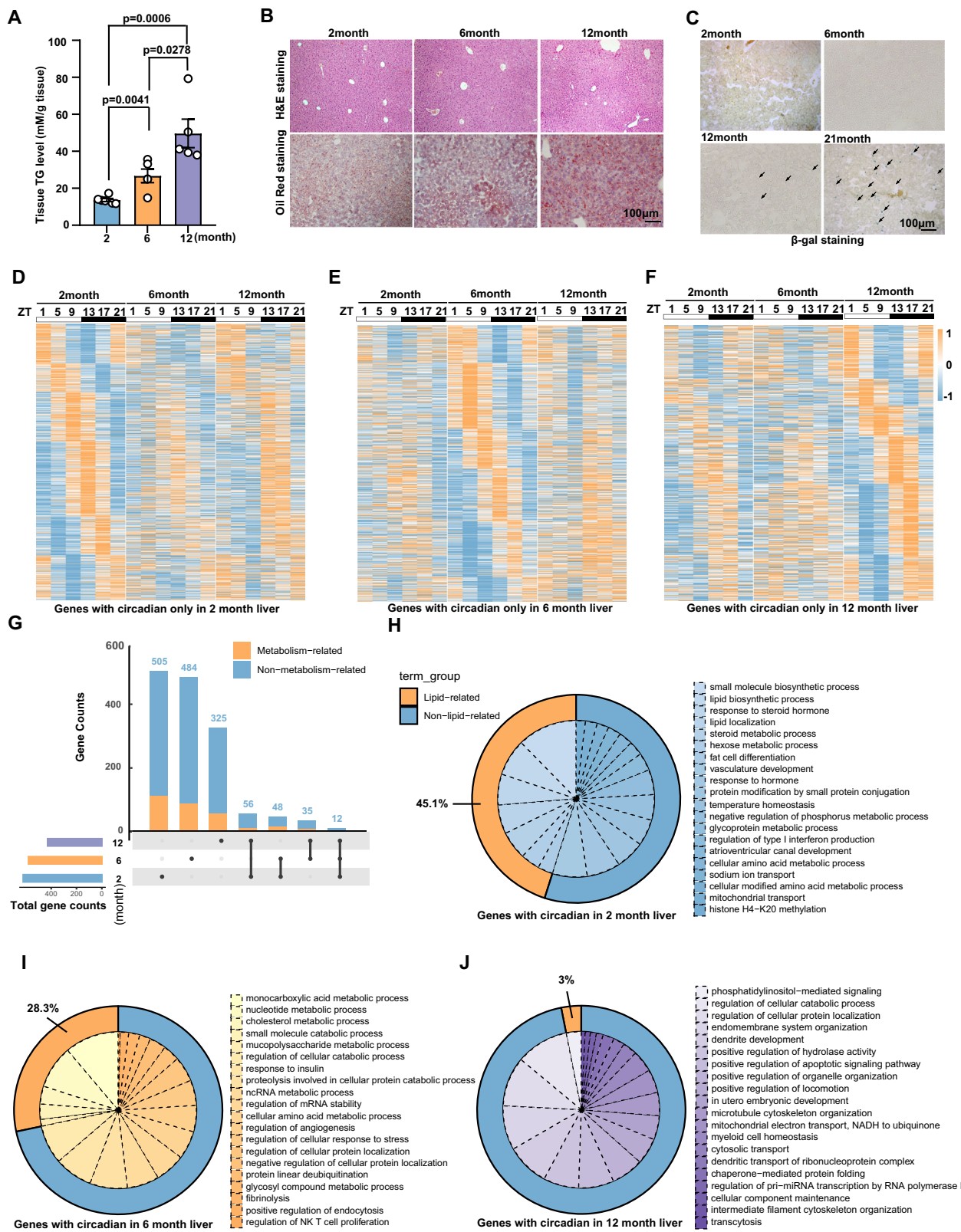

which selected via sorting the score. The results indicated that 27.2 percent of lipid-related circadian genes were regulated by Egr-1 in 2-month group, 33.8 percent of lipid-related circadian genes in 6-month group, and only 21.6 percent of genes in 12-month group (Fig. 2G–I). However, only two of these genes were consistently regulated by Egr-1 with aging (Fig. 2J). These analyses suggested that age-induced changes in Egr-1 rhythms may lead to either targeting

different lipid metabolism genes or shifting their rhythm, which in turn results in metabolism dysfunction.

### Egr-1 deletion accelerates liver age-related metabolic dysfunction

To determine the causal relationship between Egr-1 circadian rhythm and metabolic process, we generated liver-specific *Egr-1*-knockout

**Fig. 1 | The rhythmic lipid metabolism was disordered with aging. A** Liver TG levels of C57BL/6 J mice at 2 months, 6 months, and 12 months (2 months: *n* = 6; 6 months: *n* = 5; 12 months: *n* = 5 biologically independent animals); **B** H&E and Oil Red O staining of liver tissues of C57BL/6 J mice at 2 months, 6 months, and 12 months; **C** β-Gal staining of liver tissues of C57BL/6 J mice at 2 months, 6 months, 12 months, and 21 months; **D–F** Heatmap represents rhythmic genes exclusively in the livers of C57BL/6 J mice at 2 months, 6 months, and 12 months by using high-throughput RNA sequencing. The colors from blue to yellow indicate low to high gene expression levels, respectively; **G** Venn diagram displays the total number of rhythmic genes (left) and number of rhythmic non-metabolic or metabolic genes (right) in the liver; the black dots mean genes only in the indicated groups; a black line connecting black dots indicates that genes are in the connected group at the same time; **H–J** Pie charts indicate selected Top 20 biological process by using gene ontology (GO) analysis of genes circadian in 2 months, 6 months, and 12 months groups. The yellow of the outer circle means a lipid-related pathway, blue of the outer circle means a non-lipid-related pathway. Data were represented as mean ± SEM. Exact *p* values are depicted in the figure. Statistical analysis was performed using one-way ANOVA for **A**. Source data are provided as a Source Data file.

(KO) mice using *Alb-Cre* transgenic mice. We found that *Egr-1* deficiency aggravated aging-related liver lipid metabolic dysfunction. There were no obvious alterations in whole-body weight until 21 months of age, when obvious obesity was found in liver-specific *Egr-1* KO mice compared with wild-type (WT) mice (Supplementary Fig. 3A, B). There were significant decreases in liver weight of *Egr-1* KO mice aged 2-, 6-, and 12-month-old, although no significant differences were observed in 21-month-old mice (Supplementary Fig. 3C). However, the liver/body weight ratio was already decreased in the group of 2-month-old KO mice (Supplementary Fig. 3D). At the 6th month, the liver TG levels in KO mice were markedly increased, reaching the levels in 12-month-old WT mice; and then TG levels remained at the same magnitude until the 21st month, when they increased again to approximately 90 mM/g tissue (Fig. 3A). H&E staining and oil Red O staining also proved that TGs accumulated and that lipid droplet size increased in hepatocytes after *Egr-1* deletion (Fig. 3B, C). Meanwhile, the liver-free fatty acid level in KO mice aged 6-month-old were also significantly increased (Fig. 3D) and the serum-free fatty acid level were decreased (Fig. 3E). Since Egr-1 is a transcription factor, its deletion should alter the expression of downstream genes. We found that Egr-1 deficiency in the 6-month-livers of mice significantly augmented the amplitude of fatty acid uptake genes such as CD36 (Supplementary Fig. 3E), shifted the phase of fatty acid uptake gene FATP (Supplementary Fig. 3F); and suppressed the rhythm of de novo lipogenesis (Supplementary Fig. 3H–L) and TG transport-related genes (Supplementary Fig. 3R–T). The rhythmic expression of genes for fatty acid oxidation were not significantly affected by Egr-1 deletion. Thus, Egr-1 deficiency disrupted the lipid flow (flux) "in and out" balance to enable the accumulation of excessive fatty acids from 6 months onward. Moreover, Sirius Red staining and fibrosis-related gene detection showed that the fatty liver in *Egr-1*-deleted mice further developed into mild fibrosis (Fig. 3F, G). Moreover, *Egr-1* deficiency enhanced hepatocyte aging (Fig. 3H) and even slightly shortened the survival times of the mice (Fig. 3I). According to the above results, we concluded that Egr-1 could regulate metabolic dysfunction in a manner dependent on its transcriptional effects by affecting lipid metabolism.

### Transcriptomic analysis of the livers in *Egr-1*-deleted mice

To determine the underlying mechanism of Egr-1-regulated metabolic dysfunction in the liver, the WT, and *Egr-1*-KO liver samples were collected at Egr-1 highest(H) and lowest(L) zeitgeber time in mice of different ages. Liver samples were obtained at ZT13(H) and ZT17(L) in 2-month group; at ZT9(H) and ZT21(L) in 6-month group; at ZT5(H) and ZT21(L) in 12-month group; then hepatic transcriptomic analysis was performed. Heatmaps displayed significant differential gene expression patterns in each group (Fig. 4A). Venn diagrams showed the selected genes which were directly upregulated or downregulated by Egr-1 at different ages (Fig. 4B and Supplementary Fig. 4A). GO enrichment analysis of these altered genes were accomplished. Interestingly, the processes, which were selected via significantly enriched associated with lipid, mostly have a negative function of liver lipid metabolism in 2-month group, such as negative regulation of lipid storage, negative regulation of lipid localization, and negative regulation of lipid biosynthetic process. However, in 12-month group, that of

GO term mainly involved in the positive function of lipid metabolism, including regulation of lipid localization, lipid transport, positive regulation of lipid localization, and so on (Fig. 4C). Given that the shifted phase of Egr-1 circadian rhythm accelerates liver age-related metabolic dysfunction, more precise genes were obtained by overlapping genes that indirect regulated by Egr-1 in 2-month group and genes that were directly regulated by the rhythm of Egr-1 in 12 month group. Further analysis these genes showed that four upregulated genes (*Cyp2u1, Enho, Apoa1, Smim22*) were related to lipid metabolism and only one downregulated gene *Cidea* was relevant to lipid metabolism (Fig. 4D and Supplementary Fig. 4B, C).

Moreover, Euclidean distances were calculated among all samples at different ages. We found WT_6m_H and KO_6m_H groups showed the biggest difference (Fig. 4E). Phenotype differences were also observed from the 6th month; thus, we compared the transcriptomes between the WT_H and *Egr-1*-KO_H groups at 6 months, screening 613 genes with significant changes. Among these genes, 309 genes showed significant increases, such as *Mup17, Mup19, Cidea*, and *Gapdh*, while 304 genes showed significant decreases, such as the transcription factors *Egr-1, Egr-2, Myc*, and *Atf3* (Fig. 4F). We identified ten biological processes terms, such as the regulation of lipid catabolic process term and the lipid droplet organization term, that were significantly enriched (Fig. 4G). Among the identified enriched genes, we found that Cidea (Cell Death Inducing DFFA Like Effector A) participated in four of ten related metabolic processes, such as lipid localization, TG sequestration, and lipid catabolic process regulation (Fig. 4H). However, Egr-1 was not found in the list of genes related to these ten processes. As an early-primary response transcription factor, Egr-1 functions via its downstream secondary response genes, such as Tnf[29]. Importantly, we also isolated Cidea via overlapping 2-month group and 12-month group in Fig. 4D. Thus, Cidea might also be an Egr-1 target gene. Verification of the relative fragments per kilobase of transcript per million mapped reads (FPKM) values of Cidea showed that the expression of the gene significantly increased with age beginning in the 6th month and that *Egr-1* deletion led to an even more significant increase (Fig. 4I). Thus, we hypothesized that Egr-1 might regulate liver metabolic dysfunction through target genes such as Cidea.

### Cidea mediated the Egr-1-induced liver metabolic dysfunction

Cidea is a member of the Cide family that promotes lipid turnover and lipid droplet fusion[30–32]. To verify the RNA-seq results, we detected the expression levels of Cidea in liver samples. We found that both the mRNA (Fig. 5A) and protein levels (Fig. 5B) increased with age, and *Egr-1* deficiency augmented the elevations (Fig. 5A, B and Supplementary Fig. 5A). The expression enhancement was further confirmed in isolated primary hepatocytes, indicating that Cidea protein level was indeed augmented after *Egr-1* deletion in 6-month-old mice (Fig. 5C and Supplementary Fig. 5B). Oil Red O staining and BODIPY fluorescence staining indicated that the lipid droplet size was increased after *Egr-1* deletion in hepatocytes (Fig. 5D, E). Immunostaining also showed that the Cidea protein is highly expressed in *Egr-1* LKO hepatocytes (Fig. 5E). However, we did not detect the Cidea localized in larger lipid droplet surface, possibly because the Cidea protein is only transiently associated with lipid droplets. Further examination revealed that the

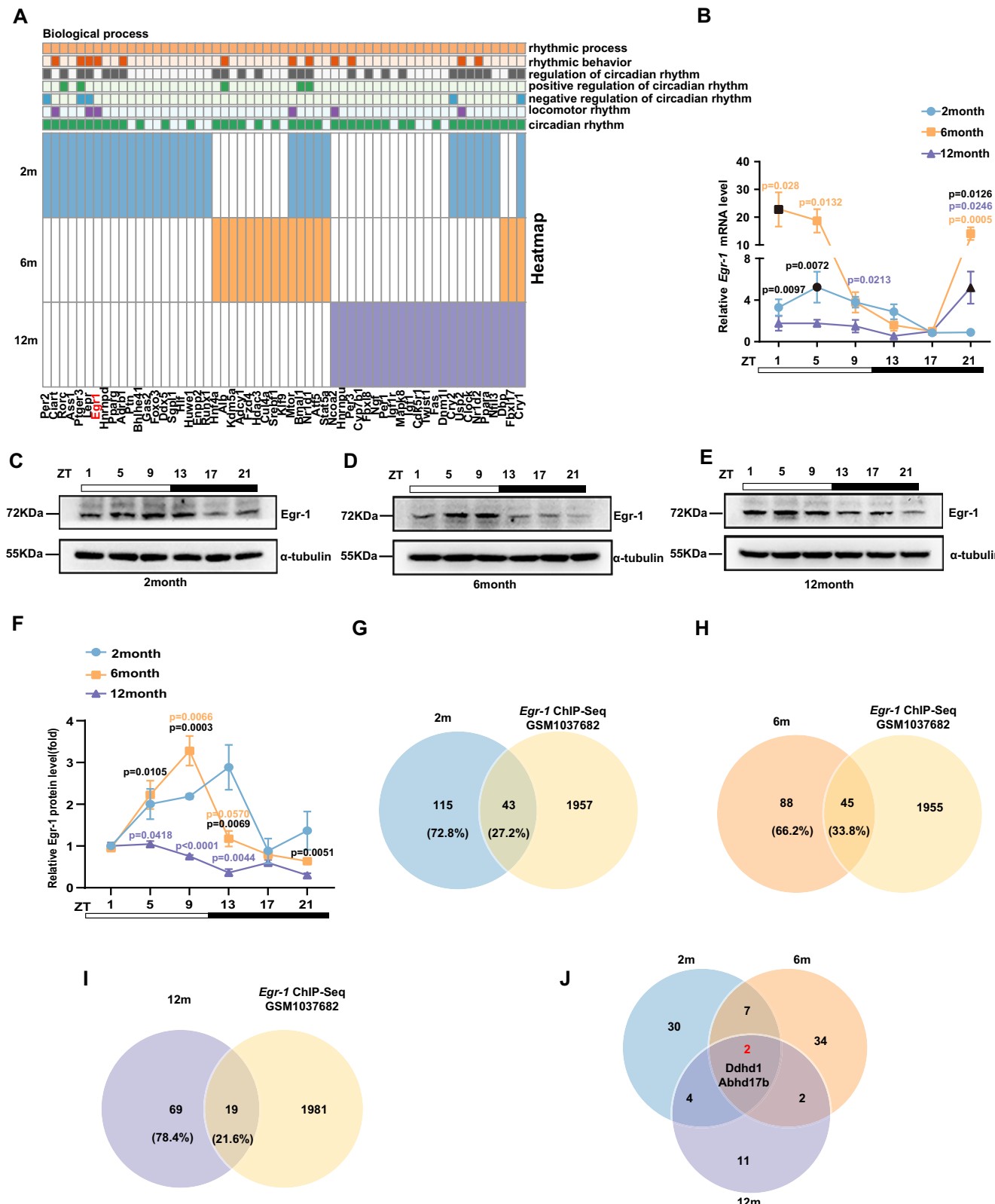

regulatory effect of Egr-1 on lipid droplets was dependent on Cidea. When we knocked down *Cidea* in primary hepatocytes, the TG level in primary hepatocytes of *Egr-1* LKO mice was dramatically decreased; the age-related increase in lipid droplet size induced by *Egr-1* LKO was blocked (Fig. 5F, G). Egr-1 overexpression decreased lipid droplet size and TG level, which was reversed by *Cidea* overexpression (Fig. 5H, I). While dnEgr-1 overexpression, which inhibits Egr-1 transcriptional

activity, increased TG level, but blocked with Cidea knocked down (Supplementary Fig. 5D). The accumulation of TG mainly depended on a fatty acid pool, which is 60% from fatty acids uptake and 25% from DNL[33]. We detected the fatty acid uptake and found that Cidea did not mediate the fatty acid uptake due to Egr-1. When overexpressing Cidea, the decrease of fatty acid uptake by overexpressing Egr-1 was not rescued (Fig. 5J).

**Fig. 2 | The phase of the Egr-1 circadian rhythm in the liver moves forward with aging. A** Heatmap represents rhythmic clock and clock-controlled genes exclusively in the livers of C57BL/6 mice at 2 months, 6 months, and 12 months; **B** mRNA expression of *Egr-1* at the indicated time points in the livers of C57BL/6 J mice at 2 months, 6 months, and 12 months (2 month ZT1: *n* = 4, ZT5,9,21: *n* = 5, ZT13,17:*n* = 6; 6 month ZT1,9: *n* = 5, ZT5,13: *n* = 4, ZT17,21:*n* = 6; 12 month: *n* = 5 biologically independent animals), black dot means the peak expression of Egr-1 point at each group. **C–E** Protein expression of Egr-1 at the indicated time points in the livers of C57BL/6 J mice at 2 months, 6 months, and 12 months (*n* = 5 biologically independent animals in 2 months; *n* = 3 biologically independent animals in

6 months; *n* = 4 biologically independent animals in 12 months). **F** Quantitative analysis of the Egr-1 protein levels in **C–E**, *n* = 3 per group. **G–I** Venn diagrams representing the overlap between WT group lipid-related genes and Egr-1 ChIP-Seq (GSM1037682) genes. **J** Venn diagrams displayed the overlap among intersecting genes in **F–H**. Data were represented as mean ± SEM. Exact *p* values are depicted in the figure. Orange color *p* value means 6 months versus 2-month group; Purple color *p* value means 12 months versus 2-month group; Black color *p* value means 6 months versus 12-month group. Statistical analysis was performed using one-way ANOVA for **B** and **F**. Source data are provided as a Source Data file.

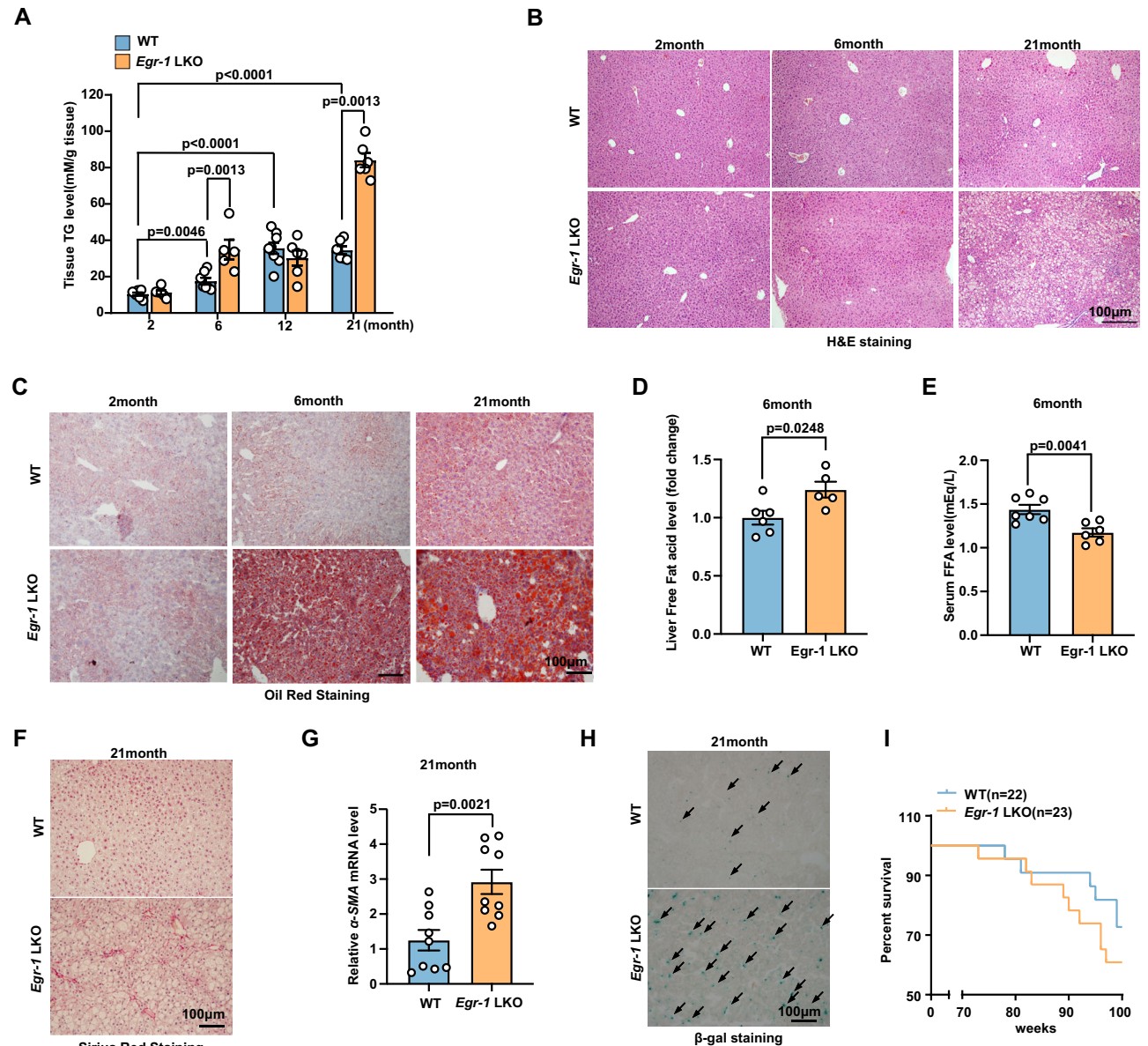

**Fig. 3 | Egr-1 deficiency accelerates liver age-related metabolic dysfunction.**
**A** Liver TG levels of WT and *Egr-1*-LKO mice at 2 months, 6 months, 12 months, and 21 months (WT: 2 months: *n* = 7; 6 months: *n* = 7; 12 months: *n* = 8; 21 months: *n* = 6; *Egr-1* LKO: 2 months: *n* = 6; 6 months: *n* = 5; 12 months: *n* = 6; 21 months: *n* = 6 biologically independent animals). **B, C** H&E staining and Oil Red O staining of WT and *Egr-1*-LKO mice at 2 months, 6 months, and 21 months of age. **D, E** Liver and serum-free fatty acid levels of WT and *Egr-1*-LKO mice at 6 months (*n* = 6 or 7 biologically independent animals in WT group and *n* = 5 or 6 biologically independent animals in *Egr-1* LKO group).

**F** Sirius Red staining of 21-month-old WT and *Egr-1*-LKO mice. **G** mRNA levels of the liver fibrosis marker a-SMA (*n* = 9 biologically independent animals). **H** β-Galactosidase staining indicates the aging process. **I** Survival curves of WT and *Egr-1*-LKO mice. Each experiment was repeated three times independently (WT: *n* = 22; *Egr-1* LKO: *n* = 23 biologically independent animals). Data were represented as mean ± SEM. Exact *p* values are depicted in the figure. Statistical analysis was performed using one-way ANOVA for **A** and unpaired two-tailed Student's *t*-test for **D, E**, and **G**. Scale bar: 100 μm. Source data are provided as a Source Data file.

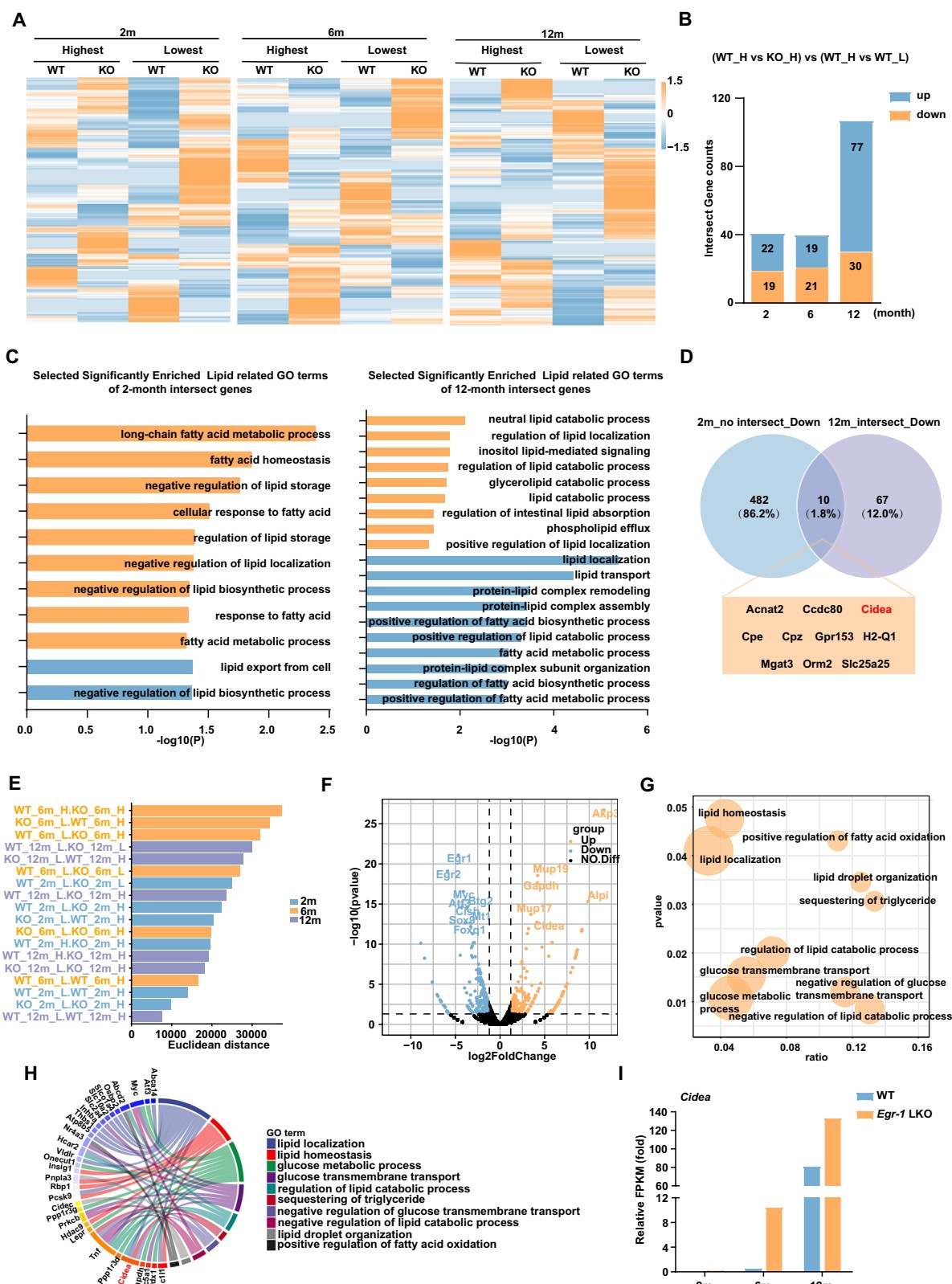

We have found that *Egr-1* deficiency in mice liver augmented the expression of fatty acid uptake gene CD36 in 2, 6, and 12 month groups (Supplementary Figs. 3E, 6A, B). Indeed, the CD36 protein level was remarkably inhibited by overexpressing Egr-1 (Supplementary Fig. 6C, D) and significantly augmented in KO mice (Supplementary Fig. 6E, F). We also predicted five putative Egr-1 binding sites in the *Cd36* promoter sequence according to the JASPAR website (Supplementary

Fig. 6G). However, we did not find the rhythmic phase of CD36 changed with age increased (Supplementary Fig. 6H). Another possibility that Egr-1 deficiency enhanced hepatic FFA level is that DNL from amino acids might also be augmented. Go enrichment showed that *Egr-1* deficiency could significantly facilitate the amino acid transport, such as amino acid transport and regulation of amino acid transmembrane transport (Supplementary Fig. 6I). Further analysis of these

**Fig. 4 | Transcriptomic analysis of the liver in Egr-1-deleted mice with age increased.** **A** Heatmap representation of the genes at Egr-1 highest and lowest zeitgeber time of mice aged 2 months, 6 months, and 12 months. The colors from blue to yellow indicate low to high gene expression levels, respectively. **B** Intersect gene counts statistics of Venn diagrams in Supplementary Fig. 4A. **C** Selected significantly enriched lipid-related GO terms of 2-month and 12-month intersect genes. The statistical test of data analysis was performed using a hypergeometric test, two-tailed, no adjustment ($p < 0.05$). **D** Venn diagrams representing the overlap between 2m_no_intersect_down and 12_intersect_down group. The lipid metabolism-related genes were labeled with red font. **E** Euclidean distance of each group difference at different ages. **F** Volcano plot displayed genes expressed in WT_H and KO_H groups in mice at the age of 6 months. The relative expression changes and significance levels are shown. The statistical test of data analysis was performed using two-tailed, no adjustment ($p < 0.05$) from the limma package in the R environment. **G** Selected significantly enriched GO terms. The x-axis and y-axis represent the enrichment and significance level, respectively, and the size of the circle represents the number of genes associated with the GO term. The statistical test of data analysis was performed using a hypergeometric test, two-tailed, no adjustment ($p < 0.05$). **H** Chord diagram revealing the enrichment levels of genes related to selected GO terms. **I** Relative FPKM values of *Cidea* in WT_H and KO_H group. We used four mice per group for the analysis. One-way ANOVA for **A** and unpaired two-tailed Student's *t*-test for **D**, **E**, and **G**.

genes indicated that most genes were related to glutamine uptake. Kcnj10 acts as a channel protein and is involved in L-glutamate import. Lpcat4 and Ggt1 could enable acyltransferase activity. Slc13a3 and Slc6a19 improve the glutamine transport across the plasma membrane (Supplementary Fig. 6J). Thus, the results indicated that *Egr-1* deletion could accelerate liver TGs accumulation by enhancing CD36 expression to facilitate fatty acid uptake, augmenting FFA synthesis from amino acids like glutamine, then enhancing Cidea expression to form large lipid droplet.

### Egr-1/BMAL1/CLOCK complex regulates the rhythm of Cidea

Considering the rhythm of CD36 and amino acid-related genes have not significantly changed with age increase, we verified the rhythmicity of Cidea and found that 2-month-old mice also displayed rhythmic *Cidea* mRNA expression in the liver that peaked at ZT13 and reached a nadir at ZT9 (Fig. 6A). Similar to that of Egr-1, the rhythmicity of Cidea also shifted with age increased, and the peak of Cidea expression in mouse livers was advanced to ZT5 at the 6th month and ZT17 at the 12th month (Fig. 6A). The rhythmic alteration of Cidea from the 6th month was further confirmed by assessment of Cidea protein expression pattern at different ages (Fig. 6B, C). When we isolated primary hepatocytes from the livers of 6-month-old mice and exposed them to horse serum shock, we observed that the amplitude of Cidea expression was dramatically elevated in KO mice (Fig. 6D and Supplementary Fig. 7A). When we overexpressed Egr-1 in wild-type hepatocytes, the amplitude of Cidea expression was considerably reduced (Fig. 6E and Supplementary Fig. 7B). All the above data indicate that Egr-1 can regulate Cidea expression robustness and rhythmicity. Thus, we analyzed the promoter region of *Cidea* and predicted two GC-rich Egr-1 binding sites and an E-box sequence region. Chromatin immunoprecipitation (ChIP) and qPCR assays demonstrated that Egr-1 could bind with all of these binding sites (Fig. 6F). A luciferase assay showed that Egr-1 binding significantly inhibited the transcription of *Cidea* (Fig. 6G), which could be rescued by mutation of Egr-1 binding sites. Interestingly, E-box mutation enhanced *Cidea* transcription to a higher degree than Egr-1 binding site mutation (Fig. 6H), which suggests that the E-box might be more important for *Cidea* expression than Egr-1 binding sites. Normally, circadian genes can bind with the E-box and regulate downstream gene expression. Thus, we co-overexpressed BMAL1/CLOCK with Egr-1 and found that E-box mutation had a stronger effect than Egr-1 mutation on BMAL1/CLOCK-controlled *Cidea* expression (Fig. 6H). A coimmunoprecipitation (Co-IP) experiment further confirmed that Egr-1 could form complexes with BMAL1/CLOCK (Fig. 6I). Therefore, we conclude that aging-related Egr-1 rhythm alterations regulate circadian Cidea expression and then affect liver TG accumulation over time. Egr-1 rhythm alteration might also result in the uncoupling of Egr-1 with BMAL1/CLOCK, which is responsible for changes in the robustness and rhythmicity of Cidea expression.

### Egr-1 phase recovery via light rescues liver metabolic dysfunction

The master clock synchronizes peripheral oscillators by imposing light-regulated rest/activity rhythms and feeding/fasting cycles[34]. We

have demonstrated that both light/dark and feeding/fasting cycles can affect the rhythm stability of liver clock oscillators in an Egr-1-dependent manner[16]. Thus, we hypothesized that restoring the phase of Egr-1 in old mice to that in young mice could delay liver lipid accumulation. To test this hypothesis, we first restricted the feeding time to the daytime. Although the rhythm of Egr-1 expression was reversed (Supplementary Fig. 8A, B), we did not detect a reduction in liver lipid accumulation at 6 months of age (Supplementary Fig. 8C). Then, we advanced the light time forward 4 h according to the phase shift at the 6th month, in which the peak of Egr-1 expression moved forward from ZT13 to ZT9 (Fig. 7A). We found that the body weights (Fig. 7B) and liver weight/body weight ratios (Fig. 7C) of phase-shift mice were significantly recovered to those of young mice. Examination of the robustness of Egr-1 and Cidea expression at ZT5 showed that the Egr-1 downregulation was attenuated at both the protein and mRNA levels (Fig. 7D, E); in addition, the upregulation of Cidea was attenuated at both the protein and mRNA levels in the transfer group at 6 months of age (Fig. 7D, F). Furthermore, quantitative detection showed that age-related TG accumulation was dramatically decreased in the light transfer group (Fig. 7G). H&E staining and Oil Red O staining also confirmed that light transfer could reverse the age-related lipid droplet phenotype (Fig. 7H). Our results indicate that changing the phase of Egr-1 and then Cidea via a light shift can ameliorate metabolic dysfunction in the liver.

All the above observations indicated that the synchronization between circadian rhythm and lipid metabolism is disrupted in old age. We suggest that age-related Egr-1 alteration acts as a master regulator of both circadian rhythms and metabolic patterns in the liver. To make our point clearer, we draw a schematic for the contribution of Egr-1 rhythm to both circadian rhythms and metabolic patterns with age increased (Fig. 8). At a young age, Egr-1 combines with circadian genes BMAL1/CLOCK to form a complex, then regulating the circadian expression of Cidea to maintain the balance of lipid metabolism. With age increased, Egr-1 rhythm alteration might result in the uncoupling of Egr-1 with both circadian genes BMAL1/CLOCK and lipid metabolic genes Cidea, which results in the decoupling of liver circadian and lipid metabolic disorders in ageing mice.

## Discussion

With age increasing, both metabolic dysfunction and circadian rhythm shifts often gradually emerge. Increasing evidence has suggested that there is a link between metabolism and the circadian rhythm[26,35]. However, the underlying mechanism that synchronizes metabolism and the circadian rhythm is still largely unknown. Here, we revealed that Egr-1, a regulator of hepatic clock circuitry, could also regulate lipid metabolism by affecting Cidea expression. Both Egr-1 and Cidea showed a rhythmic expression pattern with different phases. The peak of Egr-1 protein expression appeared at daytime near the day/night transition (ZT9-ZT13), while that of Cidea appeared in the middle of the night (ZT17) in 2-month-old mice. High expression of Cidea at nighttime is beneficial for lipid metabolism since mice eat at night. Notably, a high-fat diet (HFD) at nighttime but not daytime can protect the liver from steatosis[34]. However, the

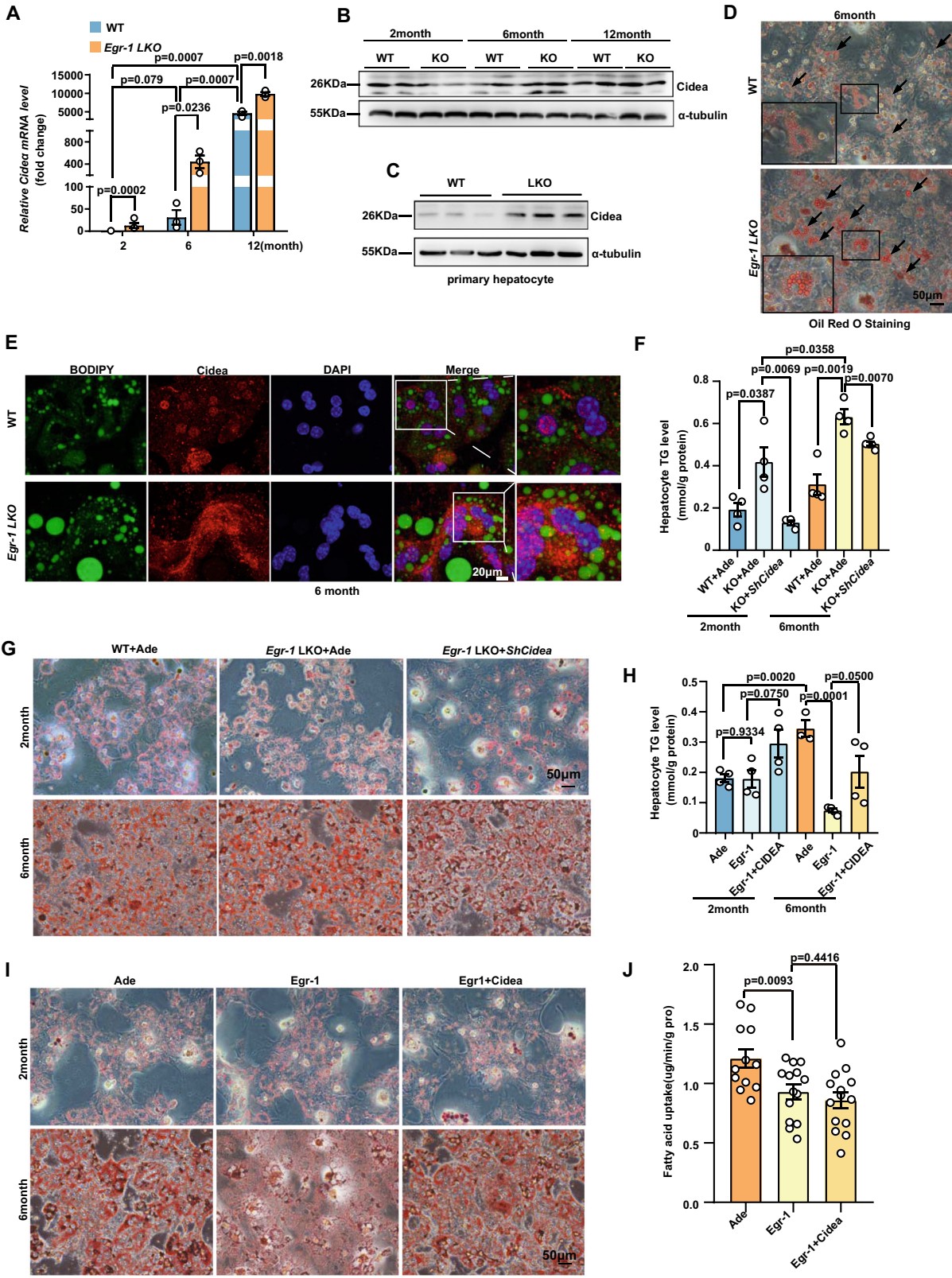

circadian expression of Egr-1 was found shifted forward with age in the current study. The Egr-1 protein expression peak shifted to ZT9 in the 6th month and ZT5 in the 12th month. This peak shift resulted in a phase shift of Cidea from ZT13 to ZT9 in the 6th and 12th months. Meanwhile, the liver showed metabolic dysfunction as a consequence of TG accumulation and lipid droplet formation increase. In our results, we found that *Egr-1* deletion results in liver TG

accumulation by enhancing CD36 expression to facilitate fatty acid uptake and Cidea expression to form larger lipid droplets. Both the robustness and rhythmicity of Cidea expression were altered under conditions of *Egr-1* deficiency. The regulatory effects of Cidea and other lipid metabolism genes on lipid metabolism during the night were therefore impaired, resulting in liver age-related metabolic dysfunction.

**Fig. 5 | Egr-1 regulates liver metabolic aging in a Cidea-dependent manner.**
**A** mRNA levels of *Cidea* in samples from RNA-seq ($n = 3$ biologically independent animals per group); **B** Protein levels of Cidea in samples from RNA-seq; **C** Protein levels of Cidea in primary hepatocytes of WT and *Egr-1* LKO mice at 6 months of age; **D** Oil Red O staining of primary hepatocytes of WT and *Egr-1* LKO mice at 6 months of age; **E** Immunofluorescence staining of primary hepatocytes of WT and *Egr-1* LKO mice at 6 months of age; **F** TG levels in WT and *Egr-1* LKO primary hepatocytes from 2-month-old and 6-month-old mice after infection with a *ShCidea* adenovirus ($n = 4$ biologically independent samples); **G** Oil Red O staining of WT and *Egr-1* LKO primary hepatocytes from 2-month-old and 6-month-old mice after infection with a *ShCidea* adenovirus; **H** Hepatocyte TG levels after transfection with a Cidea overexpression plasmid or infection with an Egr-1 overexpression adenovirus ($n = 4$ biologically independent samples); **I** Oil Red O staining after transfection with a Cidea overexpression plasmid or infection with an Egr-1 overexpression adenovirus; **J** the fatty acid uptake ratio after transfection with a Cidea overexpression plasmid or infection with an Egr-1 overexpression adenovirus ($n = 14$ biologically independent samples). Data were represented as mean ± SEM. Exact *p* values are depicted in the figure. Statistical analysis was performed using one-way ANOVA. Source data are provided as a Source Data file.

We have reported that liver Egr-1, as an early growth response factor, may rapidly mediate central signals to maintain proper oscillation of peripheral clocks[16]. After activation by master circadian signals, Egr-1 binds to the promoter of the *Per1* gene to activate its transcription, which in turn leads to repression of Per 2 and Rev-erbs. The repression of Rev-Erbs may lead to the robust oscillation of Bmal1. Then, BMAL1 binds to and activates *Egr-1* transcription. This Egr-1/Per1/BMAL1/Egr-1 feedback loop may help the liver clock keep pace with the master clock and maintain the robustness of circadian oscillation[16]. We have also reported that Egr-1 can enhance hepatic gluconeogenesis by indirectly regulating gluconeogenic gene expression via activation of C/EBPα transcription[21]. Herein, we further found that Egr-1 formed a complex with the circadian genes BMAL1/CLOCK to regulate the transcription of lipid metabolism genes such as Cidea. Our observations suggest that Egr-1 might be an important mediator between circadian and metabolic homeostasis in peripheral organs under normal conditions. However, the synchronization between metabolism and circadian rhythm is disrupted during the aging process. Circadian rhythm alterations occur in addition to decreases in Egr-1 expression, which impair both the robustness and rhythmicity of lipid metabolism genes and circadian genes during aging. Thus, metabolic homeostasis and plasticity in aging peripheral organs, such as the liver, are uncoupled from behavioral and physiological rhythms controlled by the master circadian clock that respond to light and feeding cycles, resulting in progressive metabolic dysfunction with aging.

Aging is an irresistible natural process, and the circadian rhythms of older people are normally degraded; both losses of amplitude and fragmentation of output rhythms appear in both master and peripheral circadian rhythms[36]. We have elucidated that the master clock generates circadian rhythms and drives slave oscillators in various peripheral tissues by secreting cyclical neuroendocrine signals and imposing rest/activity rhythms and feeding/fasting cycles[37–43], but we still do not know the exact factors that mediate the master clock and peripheral circadian system or the molecules that respond to these factors in peripheral organs. As we discussed above, Egr-1 can regulate both the robustness and rhythmicity of metabolic genes and circadian genes. Egr-1 can not only be transiently activated by many cytokines, growth factors, and hormones but also respond strongly to nutrient signaling under both fasting and feeding conditions. Fasting induces Egr-1 expression in the liver, which promotes hepatic gluconeogenesis by activating C/EBPα transcription[21]. After a meal, Egr-1 activity can be induced by insulin in skeletal muscle cells and inhibit insulin receptor phosphorylation, thus reducing insulin sensitivity[20]. It is reasonable to deduce that Egr-1 might be a critical liver responder that is able to integrate the oscillation of the central circadian clock and energy metabolism in peripheral organs. When the expression level and circadian rhythm of liver Egr-1 are altered during aging, metabolic homeostasis, and circadian plasticity are decoupled from the master circadian rhythm, resulting in metabolic aging in the liver.

Sleep disorders are associated with an increased risk of metabolic diseases during aging. Epidemiological studies have revealed that shift workers are more predisposed to elevated TG and high-density lipoprotein (HDL)-cholesterol levels and obesity than day workers[44]. In mammals, peripheral physiological rhythms are regulated by master circadian clocks, which respond to light and feeding cycles[34]. The timing of feeding can affect the liver biorhythm and metabolic patterns[45]. However, feeding time restriction did not ameliorate liver lipid accumulation in 6-month-old mice in the current study. Therefore, we attempted to restore the phase of Egr-1 in old mice to that in young mice via a light shift in order to examine the effect on liver metabolic aging. It has been shown that light duration and wavelength can change the central rhythm and peripheral rhythms. For example, a 6-h shift or delay in light time can cause the phase of Per1 to change in central and liver cells[46]. Filtering light below 480 nm during irradiation in a 12-h light/dark cycle promotes the expression of circadian rhythm genes in central and peripheral tissues and reduces the secretion of melatonin (a marker of aging)[47]. When we advanced the light time forward 4 h, liver metabolic aging was mostly improved. The robustness of Egr-1 and Cidea expression at ZT5 was also recovered. These results indicate that metabolic aging can be reversed by adjusting the central circadian pace according to the rhythm shift of Egr-1 at that age.

Sato et al. reported that there was no difference in the rhythm of circadian genes in young and old mice[2]. We also compared the circadian transcriptome between 2 months and 12 months of livers. There are 97 genes overlapped in total 1810 rhythmic expressed genes, which included the core clock genes with a similar rhythm between 2-month-old mice and 12-month-old mice. However, most of CCGs oscillations in 2-month-old mice were altered in 12-month-old mice, which mainly enriched in fatty acid metabolic process, lipid localization, steroid metabolic process, and cellular ketone metabolic process by Gene ontology (GO) analysis. Meanwhile, the oscillated CCGs in 12 months were not rhythmically expressed in 2-month-old mice, which enriched positive regulation of cellular protein localization, protein folding, and homeostasis of the number of cells. Thus, although the rhythms of some of the core clock genes were similar, the rhythmicity of downstream cellular and physiological functions that core clock genes regulated, such as metabolic circadian, have been altered with age increase.

In summary, we provide evidence that liver Egr-1 can act as a key responder to the master clock to integrate the central and peripheral rhythms. Liver Egr-1 also functions as an important mediator between circadian and metabolic homeostasis in peripheral organs. When the expression level and circadian rhythm of liver Egr-1 are altered during aging, the liver circadian rhythm is decoupled from the central circadian rhythm, and metabolic homeostasis and circadian plasticity are disrupted, resulting in metabolic aging in the liver. More importantly, we found that metabolic aging can be reversed by adjusting the central circadian rhythm according to the rhythm shift of Egr-1 at that age.

## Methods
### Ethical statement
This research complies with all relevant ethical regulations for the boards/committees and institutions that approved the study protocols. All mice were maintained and used in accordance with the Animal Care and Use Committee of the Model Animal Research Center of Nanjing University, Nanjing, China, using approved protocols from the institutional animal care committee (#CS20).

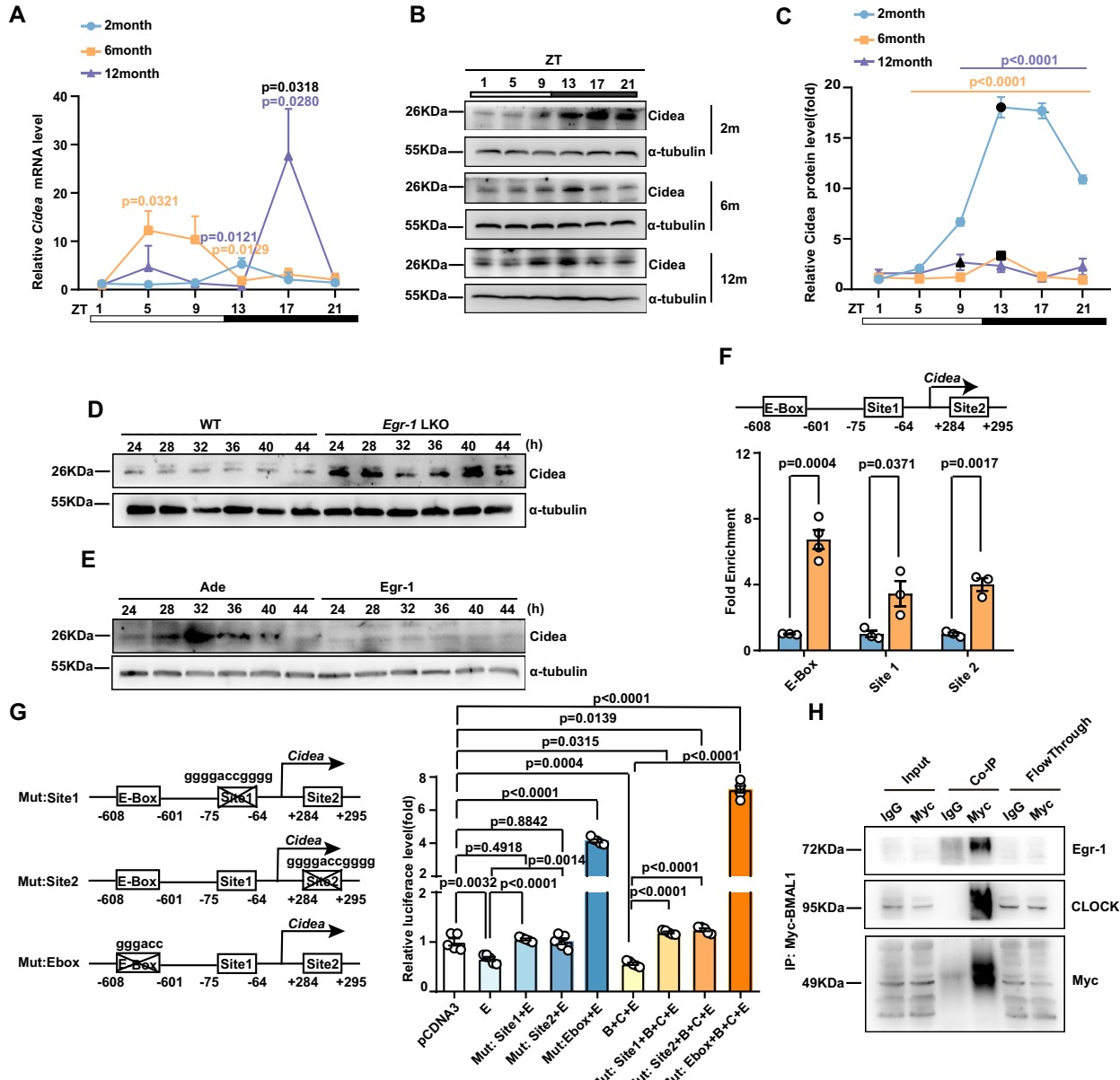

**Fig. 6 | Egr-1/BMAL1/CLOCK regulates the robustness and rhythm of Cidea by inhibiting its transcription. A** mRNA expression of *Cidea* at the indicated time points in livers from C57BL/6 J mice at 2 months, 6 months, and 12 months of age (2 months ZT1: $n = 6$, ZT5: $n = 5$, ZT9,13,17,21:$n = 4$; 6 months ZT1: $n = 4$, ZT5,13: $n = 6$, ZT9:$n = 5$, ZT17,21: $n = 4$; 12 months: $n = 4$ biologically independent animals); **B** Protein levels of Cidea at the indicated time points in livers from B6 mice at 2 months, 6 months, and 12 months of age that were entrained to a 12-h light/dark cycle. **C** Quantitative analysis of Cidea protein levels in Fig. 6B ($n = 4$ biologically independent animals per group). **D** Protein levels of Cidea in primary hepatocytes isolated from 6-month-old C57BL/6 J mice after deletion of Egr-1. **E** Protein levels of Cidea in primary hepatocytes isolated from 6-month-old C57BL/6 J mice

overexpressing Egr-1. **F** ChIP assay ($n = 3$ or 4 biologically independent samples per group); **G** Luciferase assay ($n = 5$ biologically independent samples per group). **H** HEK293T cells were co-transfected with Myc-BMAL1, Egr-1, and CLOCK expression plasmids. Co-IP of Egr-1, Myc-BMAL1, and CLOCK was performed. E means Egr-1, B means BMAL1, and C means CLOCK. Each experiment was repeated three times independently. The data represent the mean ± SEM. Exact *p* values are depicted in the figure. Orange color *p* value means 6 months versus 2 month group; Purple color *p* value means 12 months versus 2 month group; Black color *p* value means 6 months versus 12 month group. Statistical analysis was performed using one-way ANOVA. Source data are provided as a Source Data file.

## Animals

We generated mice with liver-specific KO (LKO) of *Egr-1* by crossing *Alb*-Cre transgenic mice with homozygous floxed *Egr-1* mice. Littermates were used as controls. The KO lines (strain 129) were back-crossed for a minimum of six generations to the C57BL/6 J background (*Egr-1*-loxp mouse background). The mice were housed in a controlled environment with a 12-hour/12-hour light/dark cycle at 20–24 °C and 50–65% relative humidity and were fed a chow diet ad libitum. Chow diet for reproduction and maintenance of mice were from Xietong

Shengwu, China (Reproduction: SFS9112; Maintenance: SWS9102). All the animals used in the study were male mice at 2 months, 6 months, 12 months, and 21 months of age. To analyze the lipid metabolism in Fig. 3, the liver were dissected at Egr-1 highest(H) zeitgeber time in the male mice of different ages. Liver samples were obtained at ZT13(H) in 2-month group; at ZT9(H) in 6-month group; at ZT5(H) in 12-month group, and 21-month group. To analyze gene rhythm expression, the livers of WT and *Egr-1* LKO male mice at matched ages were obtained every 6 h starting at ZT1. To analyze the function of Egr-1 in age-related

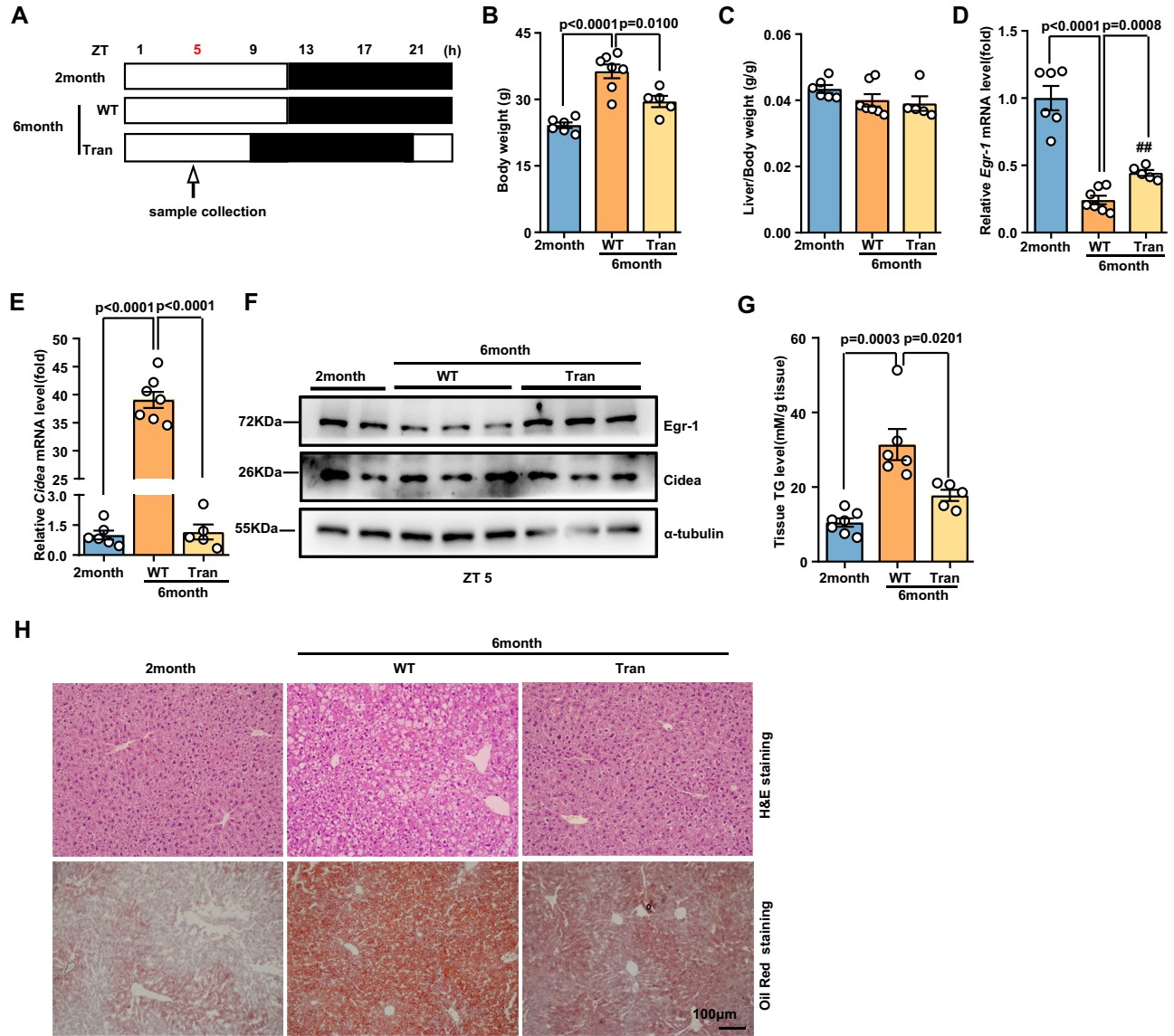

**Fig. 7 | Egr-1 phase restoration via a light shift is able to rescue liver metabolic dysfunction. A** Design of experiments in which the phase of illumination was advanced 4 h. **B**, **C** Body weights and ratios of liver weight to body weight in the groups at ZT5 (2 months: $n = 6$; 6 months: $n = 7$; 6 months Trans: $n = 5$ biologically independent animals). **D**, **E** mRNA levels of *Egr-1* and *Cidea* in the groups at ZT5 (2 months: $n = 6$; 6 months: $n = 7$; 6 months Trans: $n = 5$ biologically independent animals). **F** Protein levels of Egr-1 and Cidea in the groups at ZT5. **G** Tissue TG levels in the groups at ZT5 (2 months: $n = 7$; 6 months: $n = 6$; 6 months Trans: $n = 5$ biologically independent animals). **H** H&E staining and Oil Red O staining in the groups at ZT5. The data represent the mean ± SEM. Statistical analysis was performed using one-way ANOVA. Source data are provided as a Source Data file.

liver lipid accumulation, the light time for 6-month-old male mice was advanced by 4 h for 1 month according to the time at which the peak of Egr-1 expression moved forward from ZT13 to ZT9. The livers of male mice were dissected at ZT5 to analyze lipid metabolism. For restricted feeding, the WT and *Egr-1* LKO male mice for 6-month-old mice were fed exclusively during the daytime for 1 month. The livers of male mice were dissected every 6 h starting at ZT1 to analyze the rhythm. Mouse liver samples were obtained by cervical dislocation after anesthesia with or without light, and the mouse carcasses were treated with centralized pollution-free treatment.

In the survival experiment, the animals were euthanized when animals in a state of no anesthesia or sedation, were unable to eat or drink, and stood or extremely reluctantly stood up to 24 h. The animals were monitored once a week and once a day if the symptoms described above were present. Moribund animals were euthanized and every animal found dead or euthanized was necropsied. The criteria for

euthanasia were based on an independent assessment by the veterinarian according to the AAALAC guidelines, and the animal was represented as dead in the curve only once its condition was deemed unsuitable for continued survival. Animals in the survival curve (WT group: $n = 22$; Egr-1 LKO group: $n = 23$) were considered as censored deaths.

### Cell culture, adenovirus infection, and plasmid transfection

HEK293T (CRL-3216) cell lines were obtained from American Type Culture Collection (ATCC). Primary hepatocytes were isolated and plated at a cell density of $2 \times 10^6$ cells per well. The isolated hepatocytes and HEK293T cells were maintained in DMEM with or without Glucose containing 10% fetal bovine serum (FBS).

For adenovirus infection, primary hepatocytes were infected with adenovirus for 48 h. Control GFP- and Egr-1-expressing and dnEgr-1-expressing adenoviruses were constructed using an AdEasy adenoviral

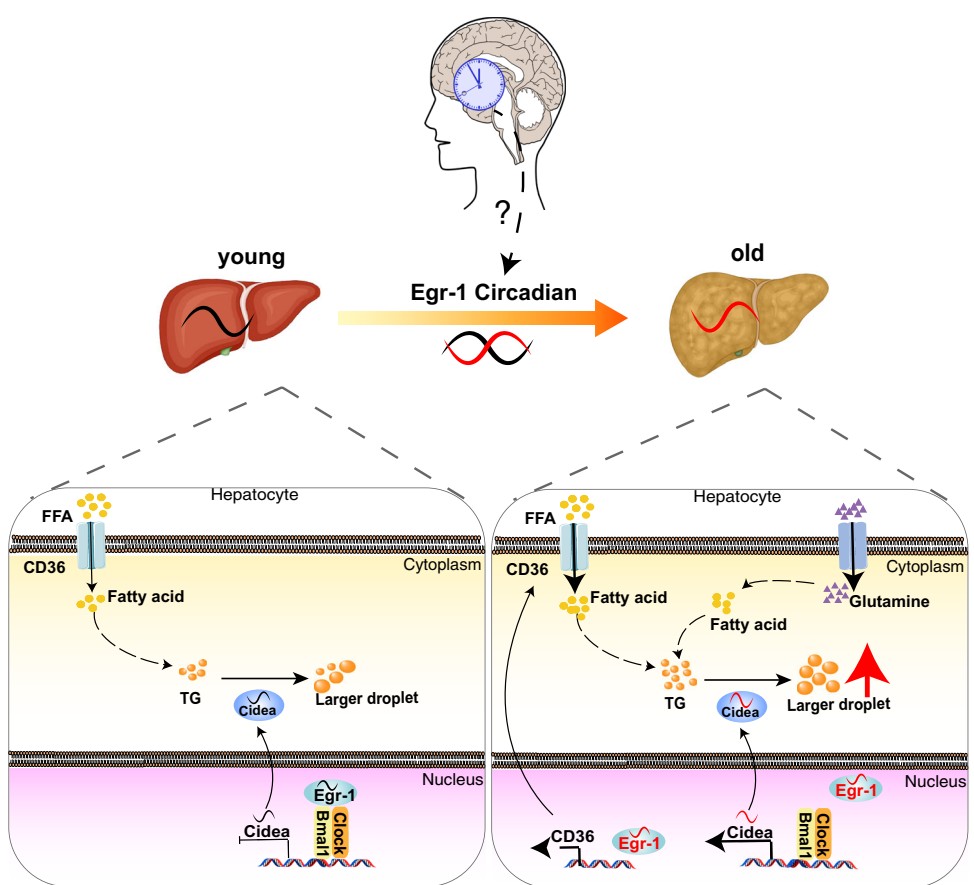

**Fig. 8 | Schematic for the contribution of Egr-1 rhythm to the correlation between circadian rhythms and metabolic patterns with age increased.** At a young age, Egr-1 combines with circadian genes BMAL1/CLOCK to form a complex, then regulates the circadian expression of Cidea to maintain the balance of lipid metabolism. With age increased, Egr-1 rhythm alteration might result in uncoupling of Egr-1 with both circadian genes BMAL1/CLOCK and lipid metabolic genes Cidea, facilitating the CD36 expression to promote the fatty acid uptake, accelerating the amino acid uptake to form more fatty acid in hepatocytes, thus, leading to the decoupling of liver circadian and the lipid metabolic disorder in ageing mice. However, how the Egr-1 rhythm responds to the master clock remains to be explored. In the work model, the elements of young and old liver were bought from the website (https://www.dreamstime.com/illust ration-non-alcoholic-fatty-liver-disease-comparison-shows-healthy-diseased-image191689868).

vector system (Stratagene, San Diego, CA, USA); then, the adenoviruses were purified by CsCl gradient centrifugation and dialyzed in PBS containing 10% glycerol. The ShCidea adenovirus was provided by Professor Li Peng (Tsinghua University, Beijing, China); the information for this adenovirus can be found in a previous study[30]. For plasmid transfection, primary hepatocytes were transfected with the indicated plasmids using Lipofectamine 2000 (Thermo Fisher Scientific, Waltham, MA, USA) according to the manufacturer's protocol. The Cidea plasmid was provided by Professor Li Peng (Tsinghua University, Beijing, China); the information for this plasmid can be found in a previous study[30]. HEK293T cells were transfected with Egr-1, BMAL1, and CLOCK plasmids or with a control PCDNA3 plasmid.

### Isolation of mouse primary hepatocytes
WT and *Egr-1* LKO mice were anesthetized via intraperitoneal injection of sodium pentobarbital (50 mg/kg). Before isolation, the perfusion buffer (#17701-038, GIBCO) with collagenase IV (C5138, Sigma) should be dissolved for 30−60 min at a temperature of 37 °C. The mouse were cleaned with 70% alcohol and made a midline incision to expose the abdominal site. Then, place the catheter into the portal vein. Start the perfusion pump at a rate of 4.5 ml/min with perfusion medium, and immediately cut SVC or IVC to let blood out for about 5 min. The liver was digested by a digest medium for 6−7 min. The liver was removed, and placed in a 100 mm plate filled with a cold 20 ml washing medium.

The cells were filtered through 100 μm filters into 50 ml centrifuge tubes and centrifuged at 50×*g* for 5 min at 4 °C. A pellet containing primary hepatocytes was resuspended in DMEM with glucose containing 10% FBS and then recentrifuged. The final pellet was resuspended in DMEM with glucose containing 10% FBS and plated at a cell density of $2 \times 10^6$ cells per six-well. Isolated primary hepatocytes were used immediately and cells can survive for ~4 days.

### mRNA and protein expression analysis and immunoprecipitation
Total RNA was extracted using TRIzol (Takara Bio, Kusatsu, Japan) according to the manufacturer's protocol. The primer sequences utilized in this study are provided in Supplementary Table 1. Q-PCR data were acquired using the Applied Biosystems Viia 7 fast Real-Time PCR system software.

For protein analysis, $2 \times 10^6$ cells or 20 mg of homogenized tissue was lysed directly in RIPA lysis buffer containing protease inhibitors (cell/tissue weight:RIPA lysis buffer volume, 1:20). Protein (50 μg per sample) was loaded onto gels. Equal amounts of protein were analyzed via western blotting with antibodies against Egr-1 (sc-101033, Santa Cruz Biotechnology, 1:500), Cidea (ab8402, Abcam, 1:1000), CD36 (sc-7309, Santa Cruz Biotechnology, 1:1000), Clock (18094-1-AP, Proteintech, 1:1000), c-Myc (10828-1-AP, Proteintech, 1:1000), β-actin (66009-1-Ig, Proteintech, 1:2000) and a-tubulin

(66031-1-Ig, Proteintech, 1:2000). Second antibodies used were: anti-rabbit-HRP (BA1054, Boster, 1:10000) and anti-mouse-HRP(BA1050, Boster, 1:10000). Western blot data were acquired digitally by Image LAB(Bio-Red) software. The quantitative analysis of western blotting bands was observed by ImageJ software 2.1.0/1.53c.

Immunoprecipitation was performed according to a standard protocol. Briefly, antibodies against Myc/IgG were added to form immune complexes with the indicated proteins in the lysates, and these complexes were immunoprecipitated using protein-A/G agarose beads (Santa Cruz Biotechnology). After several washes, the samples were boiled in 2× sample buffer and subjected to western blot analysis with Myc/Clock/Egr-1 antibodies.

## Luciferase assays
Primary hepatocytes were seeded into twelve-well dishes at a cell density of $0.8 \times 10^6$ cells per well 24 h before transfection. Egr-1, Bmal1, and Clock overexpression constructs, mCidea reporter and mutant plasmids, and internal control Renilla-luciferase plasmids were transfected. The relative luciferase activity levels were determined 48 h following transfection using a Dual-Luciferase Assay System (Promega, Madison, Wisconsin, USA) according to the manufacturer's protocol. All transfection experiments were performed in triplicate.

## H&E, Sirius Red, and β-gal staining
For histological analysis, liver tissue was collected from mice of different ages and fixed in 4% paraformaldehyde (PFA) overnight. The sections were used for H&E staining according to a standard protocol. For nuclear staining, the sections were placed in hematoxylin solution for 1 min and rinsed with distilled water.

For Sirius Red staining, paraffin sections were stained by using a Sirius Red staining Kit (Sbjbio, Nanjing, China) according to the manufacturer's protocol.

For β-gal staining, liver tissue was collected from mice of different ages and fixed in 4% PFA overnight. The sections were stained by using a β-gal staining kit (Beyotime, Shanghai, China) according to the manufacturer's protocol.

## Oil Red O tissue and hepatocyte staining
For tissue, livers were collected from mice of different ages and fixed in 4% PFA for 2 h. Then, the samples were placed in 30% sucrose solution at 4 °C overnight to remove the internal moisture, embedded in Tissue Freezing Medium (Leica, UK), and sectioned at 15-μm thickness using a Leica CM1900 Cryostat. The sections were frozen at −70 °C until staining, at which time they were air-dried, rinsed with 60% isopropanol, and stained with freshly prepared Oil Red O working solution (0.5% Oil Red O:ddH2O = 3:2) for 15 min and rinsed with 60% isopropanol.

For hepatocytes, $10^6$ cells were seeded in a 35 mm dish and treated with different experimental compounds and vehicles. The hepatocytes were fixed in 4% PFA for 15 min and then stained with freshly prepared Oil Red O working solution for the appropriate time.

## Hepatocyte BODIPY staining and immunofluorescence staining
A suitable number of cells were seeded in a 24-well plate with coverslips. The hepatocytes were fixed in 4% PFA for 20 min, incubated with 1 mg/ml BODIPY dye solution at 37 °C for 15 min, and then sealed with 50% glycerol.

For immunofluorescence staining, paraffin sections were deparaffinized, rehydrated, and boiled in citrate buffer (pH 6.0) to retrieve antigens. Afterward, the paraffin sections and frozen sections were permeabilized, blocked, and incubated with the indicated primary antibodies at 4 °C overnight. Subsequently, the sections were incubated with secondary antibodies for 1 h at room temperature. Immunofluorescence microscopy images were acquired using an Olympus

SpinSR microscope. Microscopy image analysis was performed using ImageJ software version 2.1.0/1.53c.

## Metabolic parameters
Blood samples were drawn from the retroorbital plexus. After 30 min at room temperature, the blood samples were centrifuged for 10 min at 3000×$g$ to obtain plasma, and the supernatants were collected and stored at −80 °C until analysis. Blood TG, ALT, and AST levels were determined by an automatic biochemical analyzer at Nanjing Drum Tower Hospital, the affiliated hospital of Nanjing University Medical School. The serum-free fatty acid levels were enzymatically detected with a LabAssay™ NEFA (Wako, Japan).

Liver tissue TG levels were detected using a tissue/cell TG enzymatic assay kit (Applygen, Beijing, China). Hepatocyte TG levels were detected using a TG assay kit (Jiancheng, Nanjing, China). Liver-free fatty acid levels were detected using a Free Fatty Acid Quantification Colorimetric/Fluorometric Kit (BioVison, Palo Alto, USA).

## ChIP
Primary hepatocytes were infected with an Egr-1-expressing adenovirus for 48 h. Chromatin lysates were prepared, precleared with Protein-A/G agarose beads, and immunoprecipitated with antibodies against Egr-1 or control mouse IgG. The beads were extensively washed before reverse-crosslinking. DNA was purified using a PCR purification kit (Qiagen) and subsequently analyzed by real-time PCR.

## RNA extraction, library preparation, and sequencing
The livers of C57BL/6 J mice at 2 months, 6 months, and 12 months of age were sacrificed every 4 h starting at ZT1 over the circadian cycle. The livers of WT and Egr-1 LKO mice were selected at zeitgeber time of Egr-1 highest or lowest protein expression in 2 months (H: ZT13; L: ZT17), 6 months (H: ZT9; L: ZT21), and 12 months (H: ZT5; L:ZT21) group. All samples were submitted to Novogene for transcriptome sequencing and analysis. Total RNA was extracted according to the manufacturer's procedure. The RNA quantity and purity were determined using a NanoPhotometer spectrophotometer and a Qubit2.0 Fluorometer. mRNAs were enriched with poly-T oligo-attached magnetic beads using NEBNext First Strand Synthesis Reaction Buffer (5×) and then fragmented into small pieces. These cleaved RNA fragments were reverse-transcribed to create a cDNA library using Illumina's NEBNext® Ultra™ RNA Library Prep Kit; the average insert size for the paired-end libraries was 300 bp (±50 bp). The library was sequenced on an Illumina HiSeq platform.

## RNA-seq data analysis
After sequencing and removal of low-quality reads that contained adapter contamination, low-quality bases, and undetermined bases, the sequenced reads were aligned to the mouse genome using HISAT2. Feature Counts was used to determine the mRNA expression levels by calculating the FPKM values. The expression patterns of all the genes were characterized with the TCseq package. RNA-seq data were analyzed using edgeR software version 3.38.1.

## Differential gene expression analysis
The differentially expressed genes (DEGs) were identified with edgeR software. We chose the DEGs for which the absolute value of the log (fold change) was >1.2 and the corresponding false discovery rate (FDR) was <0.05. GO annotation and enrichment of the DEGs were performed with the clusterProfiler package.

## Statistical analysis
The data are presented as the mean ± SEM from at least three independent experiments. Statistical analyses were performed using Prism 8 software (GraphPad Software, Inc., San Diego, CA, USA). The data were assessed for normal distributions with the Shapiro-Wilk test. If

the normality test was passed, statistical analyses were performed using two-tailed unpaired Student's *t*-test (two-group comparison) or one-way ANOVA followed by multiple comparisons with the LSD post hoc test (more than two groups). $P < 0.05$ was considered to indicate statistical significance.

## Reporting summary

Further information on research design is available in the Nature Portfolio Reporting Summary linked to this article.

## Data availability

All transcriptome sequencing data that support the findings of this research have been deposited in the Gene Expression Omnibus (GEO) and are accessible through the GEO accession number GSE195456. All other data generated or analysed during this study are included in this published article (and its supplementary information files). Source data are provided with this paper.

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

## Acknowledgements

This work was supported by the National Natural Science Foundation of China (91857109 to C.-J.L., 32071142 to B.X., and 31771572 to B.X.) and the Natural Science Foundation of the Jiangsu Higher Education Institutions of China (20KJB180004 to J.W.), Science and Technology Development Fund of Nanjing Medical University—General Project (NMUB2019079 to J.W.), and the Social Development Fund of Jiangsu Province (BK20191356 to B.X.). We thank Peng Li (Institute of Metabolism & Integrative Biology (IMIB), Fudan University) for providing their Cidea plasmid and ShCidea adenovirus.

## Author contributions

C.-J.L. conceptualized and designed the study. J.W., D.B., P.L. Y.Z., L.F., and B.X. coordinated the experimental planning and execution. J.W., D.B., D.S., D.C., T.Z., W.T., and M.Z. were responsible for mouse experiments. J.W., D.B., and H.W. performed the data preprocessing and statistical analysis. J.W. drafted the manuscript and produced the figures.

## Competing interests

The authors declare no competing interests.
