## [Peer Review File · Nature Communications]

The rhythmic coupling of Egr-1 and Cidea regulates age-related metabolic dysfunction in the liver of male miceREVIEWER COMMENTS

Reviewer #1 (Remarks to the Author):

The manuscript by Wu et al reports on phenotype of liver-specific Egr-1 deficient mice. The authors correlated this phenotype with changes in the expression of some of fatty acid metabolism enzymes and proposed that Cidea play some role in the process. They hypothesized that a phase shift in Erg-1 expression, observed with age, might contribute to metabolic aging. Finding on accelerated development of metabolic phenotype upon Erg-1 deficiency is interesting and reversion of metabolic phenotype in wild type mice by light phase shift is highly surprising in context of existing paradigm on the role of the circadian clock and rhythms in metabolism. The study has multiple significant weaknesses, there are issues with research design, control of some experiments and data interpretation.

Major issues.

The title is misleading, the study does not address aging. Most of the experiments in the study were done with 6 months and 12 months of age mice compared with 2 months of age mice. C57B6 mice have median lifespan about 28 months, therefore even 12-month old mice are not aged mice. 2 months of age mice are still growing mice, they are not completely adult yet, 6 months of age mice are adult, 12 months of age mice are rather early middle age.

The level of TG in the liver is significantly increased between 2 and 6 months of age in wild type mice according to Figure 1D and Figure 7H, but there is no significant increase according to Figure 3A. Which one is correct? Different units for TG were used in Figure 1D, Figure 3A and Figure 7H, what was a rationale for using different units.

Figure 2D – F was any statistical analysis performed? Was Erg-1 mRNA expression rhythmic in the liver of 2 months of age mice.

Figure 1. According to the author previous work Erg-1 is an important regulator of circadian rhythms in the liver, according to this study the expression of Erg-1 is shifted with age. At the same time, transcriptome analysis was performed only at one time point, which is not sufficient for the circadian study. Many other measurements were also done only at one time of the day.

Supplementary Figure 1. The study reported on a phase shift and increased amplitude for several circadian clock genes with age in the liver. There are several recent reports on clock gene expression in the liver of aged mice, for example by Sato et al, but no significant phase shift or an increase in the amplitude of clock gene expression rhythms were observed. The authors need to discuss it. Plus, what is an interpretation for metabolic impact of the increased amplitude of clock gene expression? According to several outstanding studies of Panda and coworkers, the increased circadian rhythms improve the metabolism, while in the current study the authors provide evidence to opposite, an increased

amplitude of rhythms results in impaired fat metabolism.

The data on survival in figure 3G are interesting, was any statistical analysis performed?

Supplementary Figure 3E-H. The expression of mRNA for metabolic enzymes, many of which are highly rhythmic, was done only at one time of the day, which has very little sense in the study that proposed circadian rhythms phase shift as mechanism of metabolic aging.

Figure 4. Again only one time point was selected for transcriptome.

Figure 5. How was TG accumulation experiment done? In culture, cells have to build TG from glucose they will take from the media. According to the data in Figure 3, Erg-1 deficiency did not impact the expression of fatty acid biosynthesis enzymes, the major effect was on fatty acid transport proteins. If fatty acid biosynthesis was not affected by the Erg-1 deficiency how can the increased accumulation of TG in Erg-1 negative cells be explained?

There are not sufficient details on the experimental design for phase 4-hour phase advance light shift experiments in Figure 7. Were animals shifted every day on 4 hours, were they shifted once? For how long were the animals on new light schedule before the analysis?

Minors

There is a significant variability in intensity of Erg-1 bands in Figure 2A western blotting, but error bars in Supplementary Figure 2 are small.

If Erg-1 suppresses Cidea expression on the level of transcription one might expect inverse correlation between Erg-1 protein level and Cidea mRNA expression. According to Figure 6A and Supplementary Figure 2D-F.

Figure 6H. What was an impact of CLOCK and BMAL1 expression on the Cidea promoter reporter? If Erg-1 suppresses the promoter through E box in CLOCK and BMAL1 dependent manner, one might expect that the mutation shall result in the loss of the effect, surprisingly, it resulted in an induction. What the interpretation.

Figure 2D-F can be combined as one panel, the same is true for Figure 6A-C.

Reviewer #2 (Remarks to the Author):

In this paper the authors describe how the circadian rhythm in the liver changes with age and this is a driver of lipid accumulation. The authors demonstrate alterations in molecular regulators of the circadian clock with ageing. Furthermore Erg-1 circadian levels change with ageing. Evidence is presented that Erg-1 is a repressor of CIDEA expression and in older mice the elevated CIDEA is associated with increased lipid storage.

The title of Figure 1 is not appropriate as peripheral circadian rhythm is not covered in this figure. It is mostly focused on how lipid metabolism gene networks are affected by aging. I would assume if clock genes were identified as differentially expressed from the analysis it would have been highlighted by the authors. This of course does not rule out that circadian pattern of clock gene expression is perturbed with ageing.

The high induction of CIDEA of 6 month Erg-1 KO mice is convincing, as is the increase in TG in the liver of these mice. However, it is unlikely that CIDEA alone is causing this and it would be more appropriate to include the data for the 6 month mice analysing FA uptake, FA synthesis, TG transport and FA oxidation from the supplementary figure the main figure.

I am concerned that the immunofluorescent imaging does not show the predicted localisation of CIDEA. Previous data strongly indicates that CIDEA should be surrounding LDs or at sites joining two LDs. The data in figure 5E do not show this at all. My examination of the images indicate that they show the CIDEA as localised to the plasma membrane in some cells and perhaps other structures within the cells. However, it is not present on LDs (or nuclei). This point of concern requires comment by the authors and likely further validation that the antibody used is actually detecting CIDEA.

The data showing Erg-1 binding to the CIDEA promoter and repression of promoter activity is convincing. Furthermore, shifting the phase of Erg-1 in 6 month old mice profoundly reduced CIDEA expression and lipid accumulation in the liver.

Overall, this is novel and interesting work. However, my concern is that it is unlikely that elevated CIDEA expression alone can account for the increase in TG liver due to Erg-1 in ageing mice. The data in the supplementary figures may point to increased uptake of fatty acid along with elevated levels of beta-oxidation. These processes should be investigated or at least discussed in further detail. The authors could undertake assays of fatty acid uptake.

Additional comments:

The first sentence of the discussion does not make sense. It is likely the authors mean "with increasing age.....". Also, the title is clumsy with the use of the word rhythm. Rhythmic would be a better word.

REVIEWER COMMENTS

Reviewer #1 (Remarks to the Author):

The manuscript by Wu et al reports on phenotype of liver-specific Egr-1 deficient mice. The authors correlated this phenotype with changes in the expression of some of fatty acid metabolism enzymes and proposed that Cidea play some role in the process. They hypothesized that a phase shift in Egr-1 expression, observed with age, might contribute to metabolic aging. Finding on accelerated development of metabolic phenotype upon Egr-1 deficiency is interesting and reversion of metabolic phenotype in wild type mice by light phase shift is highly surprising in context of existing paradigm on the role of the circadian clock and rhythms in metabolism. The study has multiple significant weaknesses, there are issues with research design, control of some experiments and data interpretation.

Major issues.

1.1 The title is misleading, the study does not address aging. Most of the experiments in the study were done with 6 months and 12 months of age mice compared with 2 months of age mice. C57B6 mice have median lifespan about 28 months, therefore even 12-month old mice are not aged mice. 2 months of age mice are still growing mice, they are not completely adult yet, 6 months of age mice are adult, 12 months of age mice are rather early middle age.

Response: Thank you for your kindly suggestion. We have changed the title to “The rhythmic coupling of EGR-1 and CIDEA regulates age-related metabolic dysfunction in the liver of mice”

1.2 The level of TG in the liver is significantly increased between 2 and 6 months of age in wild type mice according to Figure 1D and Figure 7H, but there is no significant increase according to Figure 3A. Which one is correct? Different units for TG were used in Figure 1D, Figure 3A and Figure 7H, what was a rationale for using different units.

Response: We appreciate for the reviewer’s kindly suggestion. This is our mistake for ignoring the statistical analysis of WT group in manuscript Figure 3A. We have analyzed the level of TG between 2 and 6 months of age in Figure 3A and the p value is less than

0.01(Response Figure 1B). Thus, the result was consistent with another two figures. We have also unified the concentration units to mM/g tissue (Response Figure 1A-C).

Response Figure 1 (A to Revised Figure 1A; B to Revised Figure 3A; C to Revised Figure 7G) Tissue TG level in each group. Data are represented as mean \pm SEM. *, # P < 0.05, **, ## P < 0.01, ***, ### P < 0.001. * means each group versus WT of 2month group, # means KO group versus WT group or Tran group versus WT group at 6 months; unpaired t test.

1.3 Figure 2D – F was any statistical analysis performed? Was Erg-1 mRNA expression rhythmic in the liver of 2 months of age mice.

Response: Thanks for your critical suggestions. To calculate the *Egr-1* mRNA rhythm in the liver of 2 months of age mice, we double checked it with more mice (n=7). The results showed that the *Egr-1* mRNA expression was rhythmic in 2-month-old mice. We have combined the Figure 2D-F as one panel in manuscript Figure 2A and compared together. The statistical analysis was also performed (Response Figure 2A). The mRNA level of EGR-1 was highest expressed at ZT5 at 2-month group (consistent with our previous results in *Scientific Reports*, 2015, Response Figure 2B).

Response Figure 2 (A to Revised Figure 2A) The rhythmic phase of EGR-1 in the liver shifts forward with aging. A. The mRNA expression of *Egr-1* at the indicated time points in the livers of C57BL/6 mice at 2 months, 6 months, and 12 months (n=6~8 per group). B. the mRNA of *Egr-1* in *Scientific Reports*, 2015. Data are represented as mean \pm SEM. *, #, \$ P < 0.05, **, ### P < 0.01, 12month group and 6month group versus 2month group at the indicated time points, unpaired t test. Black dots mean the *Egr-1* highest expression time point at each group.

1.4 Figure 1. According to the author previous work Erg-1 is an important regulator of circadian rhythms in the liver, according to this study the expression of Erg-1 is shifted with age. At the same time, transcriptome analysis was performed only at one time point, which is not sufficient for the circadian study. Many other measurements were also done only at one time of the day.

Response: Thanks for your insightful suggestion. We have supplemented new RNA-seq data at different zeitgeber time (every 4 hours started at zeitgeber time (ZT) 1 over the circadian cycle) and in different age (2, 6 and 12 months) (Response Figure 3A-C). Then,

we compared the rhythmic changes of metabolic gene expression of different age and replaced the old Figure 1. The new data indicated that the rhythmic lipid metabolism is disrupted in the liver (Response Figure 3D-F) and that the peripheral circadian system is shifted with aging (Response Figure 3G).

Response Figure 3 (A-F to Revised Figure 1D-F and 1H-J; G to Supplemental Figure 1B) The rhythmic lipid metabolism and peripheral circadian system were disordered with aging.

A-C. Heatmap represents rhythmic genes exclusively in the livers of C57BL/6 mice at 2 months, 6 months, and 12 months by using high-throughput RNA sequencing. The colors from blue to yellow indicate low to high gene expression levels, respectively. D-F. Pie charts indicate selected Top 20 biological process by

using gene ontology (GO) analysis of genes circadian in 2 months, 6 months, and 12 months groups. The yellow of outer circle means lipid-related pathway, blue of outer circle means non-lipid-related pathway. G. Heatmap represents rhythmic clock and clock-controlled genes exclusively in the livers of C57BL/6 mice at 2 months, 6 months, and 12 months.

1.5 Supplementary Figure 1. The study reported on a phase shift and increased amplitude for several circadian clock genes with age in the liver. There are several recent reports on clock gene expression in the liver of aged mice, for example by Sato et al, but no significant phase shift or an increase in the amplitude of clock gene expression rhythms were observed. The authors need to discuss it.

Response: Thanks for your critical suggestion. The difference between Sato's paper and our results are because of the age of the mice we used. We showed the clock genes expression in the liver of mice aged 2month, 6month and 12month, while Sato's work compared that in the liver of mice aged 12month and 20month. Through comparing our 12month data and Sato's 12month data, we could find that the rhythmic and amplitudes of clock genes expression is pretty similar (Response Figure 4). We have found that the TG accumulation would reach a platform after 12month in wildtype mice (Figure 3A; Response Figure 1B), thus the changes of clock genes expression may also tend to be stable after 12month. We have discussed this content in Result 1 part labeled by red font (Page 5, Line 107- 114).

Response Figure 4 the rhythmic and amplitudes of selected clock genes expression were similar in the liver of mice aged 12month between Sato's work and our results.

A mRNA expression of the clock genes at the indicated time points in the livers of C57BL/6 mice at 12 months (n=4~5 per group). B mRNA expression of the clock genes from Sato's work in *Cell*,2017.

1.5, Plus, what is an interpretation for metabolic impact of the increased amplitude of clock gene expression? According to several outstanding studies of Panda and coworkers, the increased circadian rhythms improve the metabolism, while in the current study the authors provide evidence to opposite, an increased amplitude of rhythms results in impaired fat metabolism.

Response: Thanks for your critical suggestions. About the difference of effect of circadian rhythms on metabolism between Panda's work and our results, we think it depended on the phase alteration with age of each circadian gene. We found not only the amplitude but also the phase of circadian genes was altered with aging (Response Figure 5A). Heatmap more vividly displayed that core clock genes and clock-controlled genes (CCGs) in circadian rhythm process had different rhythms in different age groups (Response Figure 5B). Our previous work showed that EGR-1 could be regulated by circadian rhythm genes *Bmal1/Clock* and feedback regulates the transcriptional activity of the biorhythm gene *Per1*. Thus, our opinion about fat metabolism impairment in old mice is because of the shifted rhythmic phase of circadian genes that disrupted by EGR-1 rhythm changes.

Response Figure 5 (Revised Supplemental Figure 1) the liver circadian system is shifted with aging. A. FPKM of the clock genes and clock-controlled genes at the indicated time points in the livers of C57BL/6 mice at 2 months, 6 months, and 12 months. B. Heatmap represents rhythmic clock and clock-controlled genes exclusively in the livers of C57BL/6 mice at 2 months, 6 months, and 12 months.

1.6, The data on survival in figure 3G are interesting, was any statistical analysis performed?

Response: Thanks for your critical suggestions. We did the statistical analysis and found the survival rate is slightly decrease in KO group, but there is no statistical significance. Maybe we need more mice to prove this observation.

1.7, Supplementary Figure 3E-H. The expression of mRNA for metabolic enzymes, many of which are highly rhythmic, was done only at one time of the day, which has very little sense in the study that proposed circadian rhythms phase shift as mechanism of metabolic aging.

Response: We appreciate for the reviewer's kindly suggestion. In Supplementary Figure 3E-T, we detected the rhythm of lipid metabolism related genes at ZT1,7,13,19hr of the day in WT and *Egr-1* LKO group (Response Figure 6). The results indicated that *Egr-1* deficiency in mice liver at 6-month-old augmented the amplitude of fatty acid uptake marker gene CD36, shifted the phase of FATP, and suppressed the rhythm of de novo lipogenesis (DNL) and TG transport related genes. The accumulation of TG mainly depended on fatty acid pool, which is 60% from fatty acids uptake and 25% from DNL. The increased fatty acid uptake was more conducive to accelerate TG accumulation. Thus, *Egr-1* deficiency disrupted the lipid flow (flux) "in and out" balance to enable accumulation of excessive fatty acids from 6 months onward.

Response Figure 6 (Revised Supplemental Figure 3) *Egr-1* deficiency disrupted the lipid flow (flux) "in and out" balance to enable accumulation of excessive fatty acids from 6 months onward.

WT and *Egr-1*-LKO mice at 2 months, 6 months, 12 months and 21 months were examined. mRNA expression of genes at the indicated time points related to fatty acid uptake(A-C), fatty acid synthesis(D-H), fatty acid oxidation(I-M) and TG transport (N-P) in the livers of WT and *Egr-1*-LKO mice at 6 months of age. Data are represented as mean \pm SEM. n=4-8 per group, *P < 0.05, KO group versus WT group, unpaired t test.

1.8 Figure 4. Again only one time point was selected for transcriptome.

Response: Thanks for your suggestion. We have added other point and reanalyzed the transcriptome. The WT and *Egr-1*-KO liver samples were collected at EGR-1 highest(H) and lowest(L) zeitgeber time. Liver samples were obtained at ZT13(H) and ZT17(L) in 2month group; at ZT9(H) and ZT21(L) in 6month group; at ZT5(H) and ZT21(L) in 12month group; then hepatic transcriptomic analysis was performed in Figure 4 (Response Figure 7). The results indicated that EGR-1 might regulate liver metabolic dysfunction through target genes such as CIDEA.

Response Figure 7 (Revised Figure 4A and 4I) Transcriptomic analysis of the liver in *Egr-1*-deleted mice with age increased. A. Heatmap representation of the genes at EGR-1 highest and lowest zeitgeber time of mice aged 2-month, 6-month and 12-month. The colors from blue to yellow indicate low to high gene expression levels, respectively. B. Relative FPKM values of *Cidea* in WT_H and KO_H group. We used four mice per group for the analysis. Data are represented as mean \pm SEM.

1.9. How was TG accumulation experiment done?

Response: We appreciate for the reviewer's kindly suggestion. For TG accumulation experiment, we isolated primary hepatocytes from the liver of WT or *Egr-1* LKO mice at 6-month-old and cultured overnight in DMEM containing 10% fetal bovine serum (FBS). Then, primary hepatocytes were treated with different stimulus and then collected to do TG detection.

1.10. Plus In culture, cells have to build TG from glucose they will take from the media. According to the data in Figure 3, Erg-1 deficiency did not impact the expression of fatty acid biosynthesis enzymes, the major effect was on fatty acid transport proteins. If fatty acid

biosynthesis was not affected by the Egr-1 deficiency how can the increased accumulation of TG in Egr-1 negative cells be explained?

Response: We appreciate for the reviewer's kindly suggestion. The expression metabolic enzyme data in Figure 3 was from one time of the day, we didn't find the impact of EGR-1 on the expression of fatty acid biosynthesis enzymes. Thus, according reviewer's pervious suggestion (1.7, *The expression of mRNA for metabolic enzymes, many of which are highly rhythmic, was done only at one time of the day, which has very little sense in the study that proposed circadian rhythms phase shift as mechanism of metabolic aging.*), we detected the rhythm of lipid metabolism related genes at ZT1,7,13,19 of the day in WT and *Egr-1* LKO group in 6-month group. We reanalyzed the rhythm of lipid metabolism related genes and found that *Egr-1* deficiency in mice liver at 6-month-old augmented the amplitude of fatty acid uptake gene CD36 and suppressed the rhythm of de novo lipogenesis (DNL) and TG transport related genes. The accumulation of TG mainly depended on fatty acid pool, which is 60% from fatty acids uptake and only 25% from DNL. The increased fatty acid uptake was more conducive to accelerate TG accumulation (Response Figure 8). Thus, *Egr-1* deficiency disrupted the lipid flow (flux) "in and out" balance to enable accumulation of excessive fatty acids from 6 months onward. That is why *Egr-1* deficiency enable accumulation of excessive fatty acids from 6 months onward.

Response Figure 8 the balance of lipid input and output (*Nature Endocrinology & Metabolism*, 2006)

1.11, There are not sufficient details on the experimental design for phase 4-hour phase advance light shift experiments in Figure 7. Were animals shifted every day on 4 hours, were they shifted once? For how long were the animals on new light schedule before the analysis?

Response: We appreciate for the reviewer's kindly suggestion. We have added the detail information in Animals of Materials and Methods (Page 25, Line 482-485). Detailly, to analyze the function of EGR-1 in age-related liver lipid accumulation, the light time for 6-month-old mice was advanced by 4 hours for 1 month according to the time at which the peak of EGR-1 expression moved forward from ZT13 to ZT9.

Minors

There is a significant variability in intensity of Egr-1 bands in Figure 2A western blotting, but error bars in Supplementary Figure 2 are small.

Response: Thanks for your suggestion. We have reanalyzed the EGR-1 expression between young and old group. Analyzing gene array data (GSE57809) of live young and old mice, we found that the mRNA levels of EGR-1 were markedly decreased in old mice (Response Figure 9A). We also detected the protein levels of EGR-1 at ZT5 and further confirmed their slightly decline in the livers of 21-month-old mice (Supplemental Figure 2B-C) (Response Figure 9B-C).

Response Figure 9 (Revised Supplemental Figure 2A-C) EGR-1 expression levels in the livers of C57BL/6 mice at 2 months and 21 months. A. mRNA levels of Egr-1 in young and old mice according to gene array analysis (GSE57809). B-C. Immunoblot and quantitative analysis of EGR-1 expression levels in the livers of C57BL/6 mice at 2 months and 21 months. $n \geq 3$ per group, Data are represented as mean \pm SEM. * $P < 0.05$, unpaired t test.

If Egr-1 suppresses Cidea expression on the level of transcription one might expect inverse correlation between Egr-1 protein level and Cidea mRNA expression. According to Figure 6A and Supplementary Figure 2D-F.

Response: Thanks for your suggestion. We have analyzed the correlation between EGR-1 protein level and *Cidea* mRNA expression. In 2month group, we did not find the obvious inhibition of EGR-1 on *Cidea* transcription level. With age increased, we found when EGR-1 protein expression in the liver started increased, the *Cidea* mRNA level started decreased in 6month groups. The inhibition function has become more pronounced in 12month group. Almost, the lowest EGR-1 expression ZT time was the highest *Cidea* expression point in 12month group (Response Figure 10).

Response Figure 10 (A to Revised Figure 2B; B to Revised Figure 6A) Rhythmic EGR-1 protein level inhibited the rhythmic *Cidea* mRNA level with age increased. The mRNA expression of *Egr-1*(A) and *Cidea* (B) at the indicated time points in the livers of C57BL/6 mice at 2 months, 6 months, and 12 months. Data are represented as mean \pm SEM. *, #, \$ $P < 0.05$, **, ### $P < 0.01$, 12month group and 6month group versus 2month group at the indicated time points, unpaired t test.

Figure 6H. What was an impact of CLOCK and BMAL1 expression on the Cidea promoter reporter? If Erg-1 suppresses the promoter through E box in CLOCK and BMAL1 dependent manner, one might expect that the mutation shall result in the loss of the effect, surprisingly, it resulted in an induction. What the interpretation.

Response: We appreciate for the reviewer's kindly suggestion. Firstly, from our previous results, we suggested that CLOCK and BMAL1 could form a complex with EGR-1 and suppresses the promoter through E box. Then, we initially thought that the phenomenon that the mutation resulted in an induction may be caused by the mutant sequence. We changed the E box mutant sequence but a similar phenomenon was observed (Response Figure 11C). Moreover, even we mutated both sites of Ebox, site1 and site2 simultaneously, the inhibition of EGR-1/BMAL1/CLOCK on *Cidea* transcription was not rescued (Response Figure 11D). These results indicated that E box binding site is very important for the inhibition of EGR-1/BMAL1/ CLOCK complex on *Cidea* transcription. However, we thought that there still has some spatial regulation of same chromosomes or inter-chromosome regulation, or some other unpredicted binding sites to mediate EGR-1 regulated *Cidea* transcription.

Response Figure 11 (A-B to Revised Figure 6G) EGR-1/BMAL1/CLOCK complex inhibit *Cidea* transcription level. A. the abridged general view of mutant plasmids. B-C luciferase assay. $n \geq 4$ per group, *,# $P < 0.05$, **, ## $P < 0.01$, *** $P < 0.001$, unpaired t test.

Figure 2D-F can be combined as one panel, the same is true for Figure 6A-C.

Response: Thanks for your kindly suggestion. We have combined the figures as on panel in Figure 2 and 6 (Response Figure 12).

Response Figure 12 (A to Revised Figure 2A; B to Revised Figure 6A) The mRNA expression of *Egr-1* and *Cidea* at the indicated time points in the livers of C57BL/6 mice at 2 months, 6 months, and 12 months. Data are represented as mean \pm SEM. *, #, \$ P < 0.05, **, ### P < 0.01, 12month group and 6month group versus 2month group at the indicated time points, unpaired t test. Black dots mean the *Egr-1* highest expression time point at each group.

Reviewer #2 (Remarks to the Author):

In this paper the authors describe how the circadian rhythm in the liver changes with age and this is a driver of lipid accumulation. The authors demonstrate alterations in molecular regulators of the circadian clock with ageing. Furthermore Erg-1 circadian levels change with ageing. Evidence is presented that Erg-1 is a repressor of CIDEA expression and in older mice the elevated CIDEA is associated with increased lipid storage.

2.1 The title of Figure 1 is not appropriate as peripheral circadian rhythm is not covered in this figure. It is mostly focused on how lipid metabolism gene networks are affected by aging. I would assume if clock genes were identified as differentially expressed from the analysis it would have been highlighted by the authors. This of course does not rule out that circadian pattern of clock gene expression is perturbed with ageing.

Response: thanks for your kindly suggestion. We have corrected the title of Figure 1 to “The rhythmic lipid metabolism was disordered with aging”. The analysis content of circadian rhythm change with aging was added in Supplemental Figure 1.

2.2 The high induction of CIDEA of 6-month Erg-1 KO mice is convincing, as is the increase in TG in the liver of these mice. However, it is unlikely that CIDEA alone is causing this and it would be more appropriate they include the data for the 6-month mice analyzing FA uptake, FA synthesis, TG transport and FA oxidation from the supplementary figure the main figure.

Response: We appreciate for the reviewer's kindly suggestion. We totally agree that CIDEA alone could not cause TG accumulation. Thus, we have reanalyzed the rhythm of FA uptake, FA synthesis, TG transport and FA oxidation related genes in our new transcriptive data of full circadian time series in WT and *Egr-1* KO mice. The results indicated that *Egr-1* deficiency in mice liver at 6-month-old augmented the amplitude of fatty acid uptake marker gene CD36 and suppressed the rhythm of de novo lipogenesis and TG transport related genes. The accumulation of TG mainly depended on fatty acid pool, which is 60% from fatty acids uptake and 25% from DNL. The increased fatty acid uptake was more conducive to accelerate TG accumulation (Response Figure 13). Thus, *Egr-1* deficiency may also disrupt the lipid flow (flux) "in and out" balance to enable accumulation of excessive fatty acids from 6 months onward.

Response Figure 13 (Revised Supplemental Figure 3) *Egr-1* deficiency disrupted the lipid flow (flux) "in and out" balance to enable accumulation of excessive fatty acids from 6 months onward.

WT and *Egr-1*-LKO mice at 2 months, 6 months, 12 months and 21 months were examined. MRNA expression of genes at the indicated time points related to fatty acid uptake(A-C), fatty acid synthesis(D-H), fatty acid oxidation(I-M) and TG transport (N-P) in the livers of WT and *Egr-1*-LKO mice at 6 months of age. Data are represented as mean \pm SEM. n=4-8 per group, *P < 0.05, KO group versus WT group, unpaired t test.

2.3 I am concerned that the immunofluorescent imaging does not show the predicted localisation of CIDEA. Previous data strongly indicates that CIDEA should be surrounding LDs or at sites joining two LDs. The data in figure 5E do not show this at all. My examination of the images indicate that they show the CIDEA as localised to the plasma membrane in some cells and perhaps other structures within the cells. However, it is not

present on LDs (or nuclei). This point of concern requires comment by the authors and likely further validation that the antibody used is actually detecting CIDEA.

Response: Thanks for your suggestion. We used anti-CIDEA antibody (Abcam, Ab8402) to detect it and improved the image's quality. From the image, we found CIDEA was colocalization with lipid droplets and nuclei (Response Figure 14).

Response Figure 14 (Revised Figure 5E) Immunofluorescence staining of primary hepatocytes of WT and Egr-1-LKO mice at 6 months of age.

2.4 The data showing Erg-1 binding to the CIDEA promoter and repression of promoter activity is convincing. Furthermore, shifting the phase of Erg-1 in 6-month-old mice profoundly reduced CIDEA expression and lipid accumulation in the liver.

Overall, this is novel and interesting work. However, my concern is that it is unlikely that elevated CIDEA expression alone can account for the increase in TG liver due to Egr-1 in ageing mice. The data in the supplementary figures may point to increased uptake of fatty acid along with elevated levels of beta-oxidation. These processes should be investigated or at least discussed in further detail. The authors could undertake assays of fatty acid uptake.

Response: We appreciate for the reviewer's kindly suggestion. This is a good point. according reviewer's pervious suggestion (*2.2 The high induction of CIDEA of 6-month Egr-1 KO mice is convincing, as is the increase in TG in the liver of these mice. However, it is unlikely that CIDEA alone is causing this and it would be more appropriate they include the data for the 6-month mice analyzing FA uptake, FA synthesis, TG transport and FA oxidation from the supplementary figure the main figure.*), we found that *Egr-1* deficiency in mice liver at 6-month-old augmented the amplitude of fatty acid uptake gene CD36 besides CIDEA. When we detected the fatty acid uptake ratio and found that CIDEA did not mediated the fatty acid uptake due to EGR-1. When overexpressing CIDEA, the decreased ratio of fatty acid uptake by overexpressing EGR-1 was not rescued (Response Figure 15A). We have found that *Egr-1* deficiency in mice liver augmented the amplitude of fatty acid uptake marker gene CD36 in 2month, 6month and 12month groups (Response Figure 15B-D). We supposed that EGR-1 may target regulate the CD36 expression to regulate fatty acid uptake. Indeed, the CD36 protein level was remarkedly inhibited by overexpressing EGR-1 (Response Figure 15E-F) and significantly augmented in KO mice (Response Figure 15G-

H). We also predicted five putative EGR-1 binding sites in Cd36 promoter sequence according JASPAR website (Response Figure 15I). However, we did not find the rhythmic phase of CD36 changed with age increased (Response Figure 15J). Thus, the results indicated that Egr-1 deletion can accelerate liver TG accumulation by enhancing CD36 expression to facilitate fatty acid uptake and CIDEA expression to form large lipid droplet.

Response Figure 15 (Revised Figure 5J and Supplemental Figure 5) *Egr-1* deletion can facilitate fatty acid uptake by enhancing CD36 expression. A. the fatty acid uptake ratio after transfection with a

CIDEA overexpression plasmid or infection with an EGR-1 overexpression adenovirus. B-D. mRNA expression of *CdD36* at the indicated time points in the livers of WT and *Egr-1*-LKO mice at 2months, 6 months and 12months of age. E-F. protein level and quantitative analysis of CD36 when overexpressing EGR-1. G-H. protein level and quantitative analysis of CD36 in WT and *Egr-1* LKO mice. I. five putative EGR-1 binding sites in *Cd36* promoter sequence. J. mRNA of the CD36 at the indicated time points in the livers of C57BL/6 mice at 2 months, 6 months, and 12 months. Data are represented as mean \pm SEM. * $P<0.05$, ** $P<0.01$, unpaired t test.

Additional comments:

The first sentence of the discussion does not make sense. It is likely the authors mean "with increasing age.....". Also, the title is clumsy with the use of the word rhythm.

Rhythmic would be a better word.

Response: We are sorry for this mistake and thanks for your suggestion. We have corrected the mistake in discussion (Page 22, Line 395) and title.

REVIEWER COMMENTS

Reviewer #1 (Remarks to the Author):

The authors made a substantial effort to respond to the previous round issues, Still they failed to address two major points.

First. Amplitude and phase of circadian rhythms and their impact on metabolism. The response does not help. Ironically, in response Figure 5, the rhythms in 2 months of age and 12 months of age livers are similar between each other and highly different from 12 months of age rhythms. Therefore, in my opinion, the data does not support the central statement on connection the rhythms and TG metabolism.

Second. The authors did not provide explanation on TG metabolism in hepatocytes in culture. The expression of CD36 is not relevant, the cells in culture made their fatty acids from glucose. There is no impact on glucose metabolism in KO according to the authors. Thus, cell culture data do not provide any support to in vivo data. On opposite, the cell culture data argue for different mechanism.

Both of these points are highly important to draw any conclusions.

Reviewer #2 (Remarks to the Author):

The authors have appropriately addressed the issues raised including the inclusion of new transcriptome analysis. The additional data provides more in-depth analysis of the changes in the liver with ageing. The new analysis includes the potential importance of CD36 which is now included and expanded upon in the manuscript. This clearly addresses the issues of the wider consequence of Erg-1 loss-associated increase in lipid. The authors have also amended the title to one that is more appropriate to the study. With regard to reviewer comments made in relation to the immunostaining of CIDEA, the new images do not show the protein localised around the large lipid droplets. Although the localisation is not where predicted (lipid droplet surface) the authors have appropriately not made specific claims regarding this issue. Detection of endogenous CIDEA is a challenge with immunofluorescence and it is possible that the protein is only transiently associated with lipid droplets. Therefore, I consider that the authors have addressed this issue appropriately. Overall the manuscript has been significantly improved by the changes made.

REVIEWER COMMENTS

Reviewer #1 (Remarks to the Author):

The authors made a substantial effort to respond to the previous round issues, still they failed to address two major points.

1.1 First. Amplitude and phase of circadian rhythms and their impact on metabolism. The response does not help. Ironically, in response Figure 5, the rhythms in 2 months of age and 12 months of age livers are similar between each other and highly different from 6 months of age rhythms. Therefore, in my opinion, the data does not support the central statement on connection the rhythms and TG metabolism.

Response: Thanks for your critical concerns about age-related circadian rhythms and their impact on metabolism. To confirm the similarity of rhythms in 2 months and 12 months of age livers, while they are highly different from 6 months of age that revealed by RNA sequence, we have performed Q-PCR to examine the mRNA levels of these circadian genes. The results confirmed that the phase of most circadian genes was similar in 2 months and 12 months of age livers, while the amplitude of *Bmal1* and *Clock* and *Rora* was significantly changed with age increased. And both the phase and amplitude of most circadian genes in 6 months of age livers were different from that in 2 months and 12 months of age livers (Second Response Figure 1A-I). This situation has been reported by Sato et al. in their *Cell* paper that there was no difference in the phase of circadian genes but their amplitude largely altered in the liver of young and old mice, which our results are similar with their results¹ (Second Response Figure 2). Although rhythms in 2-month-old and 12-month-old mice is similar, the impact of rhythms on lipid metabolism is totally different. Knockdown *Bmal1* or *Rora* but not *Clock* notably accelerated the hepatocytes TG accumulation at 2 month-year-old mice (Second Response Figure 1J), which is consistent with previous reports^{2, 3, 4}. On the other hand, with age increased, the regulation of these circadian genes on TG accumulation was reversed. Especially at 12month-year-old mice, knockdown of *Bmal1* or *Clock* or *Rora* both inhibited the TG accumulation (Second Response Figure 1L). It is indicated that the connection of rhythms and metabolism is transient disturbed in 6-month-old mice that Knockdown *Bmal1* or *Rora* decreased TG accumulation while *Clock* still did not affect TG accumulation (Second Response Figure 1K). It is possible the rhythm in 12-month-old mice has moved to another 24-hour rhythm thus it looks similar to that in 2-month-old mice, when other disturbing factor has involved to regulate liver lipid metabolism.

About the connection between the rhythms and TG metabolism, we are sorry that we didn't make our statement clearly. In our previous manuscript, we thought age-related Egr-1 alteration acts as a master regulator of both circadian rhythms and metabolic patterns in liver. Herein, we have changed the statement according to our new data following the reviewer's suggestion that there is another possibility that Egr-1 and BMAL1/CLOCK could form a complex to regulate circadian expression of *Cidea* to maintain the balance of lipid metabolism (Second Response Figure 3). At young age, Egr-1/BMAL1/CLOCK complex could regulate circadian expression of *Cidea* to maintain the balance of lipid metabolism. With age increased, Egr-1 rhythm alteration might result in uncoupling of Egr-1 with BMAL1/CLOCK and then *Cidea*, which results in the decoupling of liver circadian and the lipid metabolic disorder in ageing mice. Moreover, we thought that Egr-1 also

might be a critical liver responder that is able to integrate the oscillation of the central circadian clock and energy metabolism in peripheral organs, but we still do not know the exact factors.

We have changed the supplemental figure 1 to our new results and added the detail information in Result 1 Page 4 Line 109-127. We also make our statement clearer in discussion part in Page 24 Line 474-485. The working model is also showed in Figure 8.

Second Response Figure 1 (to Second Revised Supplemental Figure 1A-L) A-I. mRNA level of the clock genes and clock-controlled genes at the indicated time points in the livers of C57BL/6 mice at 2 months, 6 months, and 12 months; J-L. TG levels in WT primary hepatocytes from 2-month-old, 6-month-old and 12-month-old mice after infection with a siRNA by knocking down *Bmal1* or *Clock* or *Rora*. *, #, \$ $P < 0.05$, **, ##, \$\$\$ $P < 0.01$, ***, ###, \$\$\$ $P < 0.001$, * means 6month and 12month group versus 2month group; # means 2month and 12month group versus 6month group; \$ means 2month and 6month group versus 12month group, unpaired t test.

Cell, 2017

Second Response Figure 2 the mRNA level of selected clock genes expression in young normal diet

(YND) and old normal diet (OND) fed mice (*Cell*, 2017).

Second Response Figure 3 Schematic for the contribution of Egr-1 rhythm to the correlation between circadian rhythms and metabolic patterns with age increased.

1.2 Second. The authors did not provide explanation on TG metabolism in hepatocytes in culture. The expression of CD36 is not relevant, the cells in culture made their fatty acids from glucose. There is no impact on glucose metabolism in KO according to the authors. Thus, cell culture data do not provide any support to in vivo data. On opposite, the cell culture data argue for different mechanism.

Response: We appreciate for the reviewer's kindly suggestion. We agree that culture cells make their fatty acids from glucose and even from other resource such as amino acids. In our case, we think the fatty acids in fetal bovine serum (FBS) may be one of the resources of TAG in hepatocyte. FBS is a complex mixture of proteins, hormones, lipids and other various sized biomolecules. The

concentration of lipids in culture medium (10% FBS) would be 65.6ug/ml. In order to verifying whether FBS could provide enough lipids and eliminating the effects of glucose, we isolated primary hepatocytes from the liver of WT mice at 6-month-old and cultured in DMEM without glucose but with 10% FBS. Then, primary hepatocytes were treated with different stimulus and then collected to do TG detection. The results showed that Egr-1 overexpression decreased TG level, which was reversed by Cidea overexpression (Second Response Figure 4A); while dnEgr1 overexpression, which inhibits Egr-1 transcriptional activity, increased TG level, but blocked with Cidea knocked down (Second Response Figure 4B). Thus, our results indicated that fatty acid in FBS also contribute the TG accumulation in cultured hepatocyte along with glucose and amino acids. We have added our new data in Supplemental Figure 5 and discussed in page 16 line 312-323.

Second Response Figure 4 (to Second Revised Supplemental Figure 5C-D) Detection of primary hepatocytes TG level with different treatment in DMEM without glucose containing 10% FBS. A. Hepatocyte TG levels after transfection with a CIDEA overexpression plasmid or infection with an EGR-1 overexpression adenovirus; B. TG levels in primary hepatocytes from 6-month-old mice after infection with a ShCidea adenovirus or infection with an dnEGR-1 overexpression adenovirus. The data represent the mean \pm SEM. *, P<0.05, ** P<0.01, unpaired t test.

Reviewer #2 (Remarks to the Author):

The authors have appropriately addressed the issues raised including the inclusion of new transcriptome analysis. The additional data provides more in-depth analysis of the changes in the liver with ageing. The new analysis includes the potential importance of CD36 which is now included and expanded upon in the manuscript. This clearly addresses the issues of the wider consequence of Erg-1 loss-associated increase in lipid. The authors have also amended the title to one that is more appropriate to the study. With regard to reviewer comments made in relation to the immunostaining of CIDEA, the new images do not show the protein localised around the large lipid droplets. Although the localisation is not where predicted (lipid droplet surface) the authors

have appropriately not made specific claims regarding this issue. Detection of endogenous CIDEA is a challenge with immunofluorescence and it is possible that the protein is only transiently associated with lipid droplets.

Therefore, I consider that the authors have addressed this issue appropriately. Overall the manuscript has been significantly improved by the changes made.

Response: We appreciate for the reviewer's kindly suggestion. We agree with that the Cidea protein may be only transiently associated with lipid droplets. We have added the claims in Result 5 (Page 15, Line 304-306).

Reference

1. Sato S, *et al.* Circadian Reprogramming in the Liver Identifies Metabolic Pathways of Aging. *Cell* **170**, 664-677 e611 (2017).
2. Shimba S, *et al.* Deficient of a clock gene, brain and muscle Arnt-like protein-1 (BMAL1), induces dyslipidemia and ectopic fat formation. *PLoS One* **6**, e25231 (2011).
3. Rudic RD, *et al.* BMAL1 and CLOCK, two essential components of the circadian clock, are involved in glucose homeostasis. *PLoS Biol* **2**, e377 (2004).
4. Kim K, *et al.* RORalpha controls hepatic lipid homeostasis via negative regulation of PPARgamma transcriptional network. *Nature communications* **8**, 162 (2017).
5. Chakrabarti P, *et al.* Insulin inhibits lipolysis in adipocytes via the evolutionarily conserved mTORC1-Egr1-ATGL-mediated pathway. *Mol Cell Biol* **33**, 3659-3666 (2013).

REVIEWER COMMENTS

Reviewer #1 (Remarks to the Author):

The authors provided the response and some additional data.

The response to the first question is not on the point of the question. The statement "It is possible the rhythm in 12-month-old mice has moved to another 24-hour rhythm thus it looks similar to that in 2-month-old mice, when other disturbing factor has involved to regulate liver lipid metabolism." is highly speculative, any observations can be explain with such statements.

The response to the second request is new data in Figure 4 in the Second response. However, it does not address the concern. The concentration of glucose in FBS is about 5mM and FFAs is about 0.2 mM. The concentration of fatty acids in FBS is low compared with the concentration in adult adult serum. The transport of lipids is usually by concentration gradient. In feeding condition it is from hepatocytes and in fasting it is in. In the media with 10% FBS the concentration of FFAs is <10% of physiological concentration and about 100 times less compared with fasting. Thus, the hepatocytes need to create this gradient in order to transport FFAs from outside. They also need simultaneously to oxidize the fatty acids and to build TGs from them, which is confusing in the condition of low FFAs concentration and no glucose. Usually cells will induce autophagy at such conditions.

Reviewer #1 (Remarks to the Author):

The authors provided the response and some additional data.

1. The response to the first question is not on the point of the question. The statement "It is possible the rhythm in 12-month-old mice has moved to another 24-hour rhythm thus it looks similar to that in 2-month-old mice, when other disturbing factor has involved to regulate liver lipid metabolism." is highly speculative, any observations can be explained with such statements.

Response: We apologize for not clearly elucidating the change of circadian rhythm with age increase. At molecular level, circadian genes could be sub-grouped into core clock genes and clock-controlled genes (CCGs). The core clock genes mainly formed transcriptional feedback loops, such as Bmal1/Clock, PER and CRY proteins, driving their rhythmic oscillation during 24-hour circadian. The CCGs are regulated by the rhythm of these core clock genes directly or indirectly, then ultimately impose rhythmicity on downstream cellular and physiological functions, such as metabolic circadian. Our statement "It is possible the rhythm in 12-month-old mice has moved to another 24-hour rhythm thus it looks similar to that in 2-month-old mice, when other disturbing factor has involved to regulate liver lipid metabolism" is based on our analysis that the 24-hour rhythm of CCGs is largely altered although the that of some core clock genes is similar between 2-month-old mice and 12-month-old mice.

1. We compared the circadian transcriptome between 2 months and 12 months of livers. There are 1028 and 782 genes presented 24-hour rhythmic expression pattern, in 2-month-old mice and 12-month-old mice respectively. However, only 97 genes overlapped in total 1,810 rhythmic expressed genes, which included the core clock genes with similar rhythm between 2-month-old mice and 12-month-old mice (Third Response Figure 1A). However, most of oscillations of CCGs in 2-month-old mice lost their rhythmicity in 12-month-old mice (Third Response Figure 1B), which mainly enriched in fatty acid metabolic process, lipid localization, steroid metabolic process, and cellular ketone metabolic process by Gene ontology (GO) analysis. Meanwhile, the oscillated CCGs in 12month were not rhythmically expressed in 2-month-old mice (Third Response Figure 1C), which enriched positive regulation of cellular protein localization, protein folding and homeostasis of number of cells. Our analysis indicated that although the rhythms of core clock genes were similar, the rhythms of CCGs were totally different in the liver of 2-month-old mice and 12-month-old mice.
2. Even the rhythms of core clock genes were similar, we also noticed there was subtle different.
1), Although GO enrichment enriched the circadian rhythm related biological pathway in 2-month-old mice and 12-month-old mice, the related gene number was remarkably decreased except entrainment of circadian clock and entrainment of circadian clock by photoperiod. And some pathways even disappeared like circadian sleep/wake cycle non-REM sleep, positive regulation of circadian sleep/wake cycle, negative/positive regulation of circadian rhythm (Third Response Figure 1D). 2), The amplitude of Bmal1 and Clock and Ror α was significantly changed with age increased in mRNA level (Third Response Figure 1E-G). 3), We also found that the regulation of core circadian gene on TG accumulation is totally different between 2month- and 12month-old mice, in which knockdown of Bmal1 or Clock or Ror α could enhance TG accumulation in 2month-old mice, while decreased TG accumulation in 12-month-old mice

- (Third Response Figure 1H-I). Our data indicated that the “normal” 2-month-old relationship between core clock genes and lipid metabolism was disrupted in the liver of 12-month-old mice.
3. For the specific CCGs enriched in fatty acid metabolic process in the liver of 2-month-old mice, such as *Crat*, *Acot3*, *Cpt1a*, *Pparg*, *Fasn*, either the amplitude was enhanced (*Acot3* and *Pparg*), or the rhythm was altered (*Cpt1a*), or even loss of 24-hour rhythm (*Crat* and *Fasn*) in the liver of 12-month-old mice (Third Response Figure 1J-N). Thus, the CCGs rhythm enriched in the lipid metabolism in the liver of 2-month-old mice were altered in the liver of 12-month-old mice.

Thus, although the rhythm in 12-month-old mice moved to another 24-hour rhythm thus it looks similar to that in 2-month-old mice, the rhythm and function of its downstream CCGs like lipid metabolism have already disturbed in 12-month-old mice.

We have added these results in Result 1 part (Page 4 Line 108-123) and discussed in Discussion Part (Page 26 Line 516-527).

Third Response Figure 1 (to Third Revised Supplemental Figure 1) A.Venn diagrams representing the overlap between 2 months or 12 months group; B-C.Selected top10 significantly enriched GO terms of 2-month and 12-month intersect genes; D. Selected significantly enriched circadian rhythm related GO terms of 2-month and 12-month intersect genes; F-G, mRNA level of the core clock genes at the indicated time points in the livers of C57BL/6 mice at 2 months and 12 months. H-I. TG levels in WT primary hepatocytes from 2-month-old and 12-month-old mice after infection with a siRNA by knocking down *Bmal1* or *Clock* or *Rora*. K.FPKM values of *Crat*, *Acot3*, *Cpt1a*, *Pparg*, *Fasn* in 2 months and 12 months group. We used five mice per group for the analysis. *, #, \$ P < 0.05, **, ##, \$\$ P < 0.01, ***, ###, P < 0.001, * means 6month and 12month group versus 2month group; # means 2month and 12month group versus 6month group; \$ means 2month and 6month group versus 12month group, unpaired t test.

2.The response to the second request is new data in Figure 4 in the Second response. However, it

does not address the concern. The concentration of glucose in FBS is about 5mM and FFAs is about 0.2 mM. The concentration of fatty acids in FBS is low compared with the concentration in adult serum. The transport of lipids is usually by concentration gradient. In feeding condition, it is from hepatocytes and in fasting it is in. In the media with 10% FBS the concentration of FFAs is <10% of physiological concentration and about 100 times less compared with fasting. Thus, the hepatocytes need to create this gradient in order to transport FFAs from outside. They also need simultaneously to oxidize the fatty acids and to build TGs from them, which is confusing in the condition of low FFAs concentration and no glucose. Usually, cells will induce autophagy at such conditions.

Response: We appreciate for the reviewer's kindly suggestion. In order to confirm the FFA level of FBS we used, we detected five companies' FBS simultaneously. The results indicated that FFA levels are not similar in FBS from different companies. FFA level of FBS we used (FBSST-01033, OriCell company, from Uruguay) that is about 0.9 mM, which is similar with that of Gibco's FBS (10100147-500ml, gibco, from Australia). It is also true that there are some FBS with lower FFA levels, such as BI's 0.5 mM (04-001-1ACS, Biological Industries, from middle America); Newzerum's 0.07mM (FBS-PA500, Newzerum, from Australia) and Biochannel's 0.12mM (BC-SE-FBS, Biochannel, from China)(Third Response Figure 2A).

Then, we agree that the transport of lipids is usually by concentration gradient. We detected the FFA level of DMEM without glucose but with 10% FBS is about 0.15mM. We also checked the FFA levels in isolated primary hepatocytes from 6-month-old mice was about 0.17mM, similar with the culture medium (Third Response Figure 2B), which would decrease to almost 0.13mM or less after culture (Third Response Figure 2C). When Egr-1 activity was blocked by dnEgr-1 adenovirus, we could find that the hepatocytes' FFA level increased (Third Response Figure 2C) and the culture medium FFA level was decreased (Third Response Figure 2D). Thus, we speculated that there were two possibilities that Egr-1 down regulation could enhance FFA level and TG accumulation.

1. Egr-1 transcriptive activity blockage facilitated the fatty acid uptake gene CD36 mRNA expression (Third Response Figure 2E). Thus, the FFA level in medium decreased because of hepatocyte uptake (Third Response Figure 2D), and then the hepatocytes TG level increase (Third Response Figure 2F). The data indicated that the cultured hepatocytes could uptake fatty acids and build TGs.
2. However, given the inconspicuous decrease in FFA levels in the medium, we also suggest that amino acids may be also involved in FFA production in cultured hepatocytes. We reanalyze the transcriptome of WT and *Egr-1* LKO mice aged 6-month-old and GO enrichment showed that *Egr-1* deficiency could significantly facilitate the amino acid transport, such as amino acid transport, regulation of amino acid transmembrane transport (Third Response Figure 2G). Further analysis these genes indicated that mostly genes were related to glutamine uptake. Kcnj10 acts as a channel protein and involved in L-glutamate import. Lpcat4 and Ggt1 could enable acyltransferase activity. Slc13a3 and Slc6a19 improve the glutamine transport across the plasma membrane (Third Response Figure 2H). We speculated that hepatocyte could also use glutamine and other amino acids to synthesize FFA production.

Thus, we think that there is small gradient from outside to inside in our culture system so that cultured hepatocyte could uptake FFA from outside to build TGs although it is inconspicuous. Of course, the uptake amino acids like glutamine from medium provide another possibility that hepatocyte could synthesize FFA and build TGs.

We have added these results in Result 5 Page 16 Line 320-329.

Third Response Figure 2 (to Third Revised Supplemental Figure 6I-J) Hepatocytes could absorb the fatty acids and build TGs in the condition of low FFAs concentration and no glucose. A. FFA level of FBS in different companies(Gibco:10100147-500ml, from Australia; OriCell: FBSST-01033, from Uruguay; BI(Biological Industries): 04-001-1ACS, from middle America; Newzerum: FBS-PA500, from Australia; Biochannel: BC-SE-FBS, from China); B. FFA level of FBS we used and DMEM without glucose but with 10% FBS and primary hepatocytes before adding this conditional medium; When over-expressing dominate negative Egr-1 adenovirus in this conditional medium at 24hours: C.hepatocytes FFA level; D. FFA level of culture medium; E. Cd36 mRNA level; F.hepatocytes TG level; G. selected significantly Enriched amino acid related GO terms of changed genes between WT and *Egr-1* LKO (KO) group at 6-month-old; H. significantly changed genes in amino acid related GO terms in F. *P < 0.05, ** P < 0.01, unpaired t test.

REVIEWERS' COMMENTS

Reviewer #1 (Remarks to the Author):

The authors provide a response to my comments.